# Global Convergence of Online Optimization for Nonlinear Model Predictive Control

Sen Na

Department of Statistics
University of Chicago
Chicago, IL 60637
senna@uchicago.edu

## Abstract

We study a real-time iteration (RTI) scheme for solving online optimization problem appeared in nonlinear optimal control. The proposed RTI scheme modifies the existing RTI-based model predictive control (MPC) algorithm, by selecting the stepsize of each Newton step at each sampling time using a differentiable exact augmented Lagrangian. The scheme can adaptively select the penalty parameters of augmented Lagrangian on the fly, which are shown to be stabilized after certain time periods. We prove under generic assumptions that, by involving stepsize selection instead of always using a full Newton step (like what most of the existing RTIs do), the scheme converges globally: for *any* initial point, the KKT residuals of the subproblems converge to zero. A key step is to show that augmented Lagrangian keeps decreasing as horizon moves forward. We demonstrate the global convergence behavior of the proposed RTI scheme in a numerical experiment.

## 1 Introduction

We consider the following time-varying nonlinear optimal control problem

$$\mathcal{P}(\bar{\boldsymbol{x}}_0): \ \min_{\boldsymbol{x},\boldsymbol{u}} \ \sum_{k=0}^{N-1} g_k(\boldsymbol{x}_k, \boldsymbol{u}_k) + g_N(\boldsymbol{x}_N),$$
$$\text{s.t.} \ \ \boldsymbol{x}_{k+1} = f_k(\boldsymbol{x}_k, \boldsymbol{u}_k), \ \ k \in [N-1], \tag{1}$$
$$\boldsymbol{x}_0 = \bar{\boldsymbol{x}}_0,$$

where $\boldsymbol{x}_k \in \mathbb{R}^{n_x}$ are the state variables, $\boldsymbol{u}_k \in \mathbb{R}^{n_u}$ are the control variables, $g_k : \mathbb{R}^{n_x} \times \mathbb{R}^{n_u} \rightarrow \mathbb{R}_+$ ($g_N : \mathbb{R}^{n_x} \rightarrow \mathbb{R}_+$) are the nonnegative cost functions, $f_k : \mathbb{R}^{n_x} \times \mathbb{R}^{n_u} \rightarrow \mathbb{R}^{n_x}$ are the dynamical constraint functions, $\bar{\boldsymbol{x}}_0$ is the given initial state, and $N$ is the horizon length. We suppose that $g_k, f_k$ are twice continuously differentiable and, without loss of generality, that the origin is a steady state, i.e. $f_k(\boldsymbol{0}, \boldsymbol{0}) = \boldsymbol{0}$, and is also a stationary point with zero loss, i.e. $g_k(\boldsymbol{0}, \boldsymbol{0}) = 0$ and $\nabla g_k(\boldsymbol{0}, \boldsymbol{0}) = \boldsymbol{0}$. The well-known linear quadratic regulator (LQR) satisfies this setup and the same setup is commonly used in the literature [8]. Problem (1) is also called dynamic program in control community, which has a close relation to reinforcement learning (RL) (see [6]). The main difference is that the dynamic $f_k$ in our paper is known, which is the case in many industrial applications and model-based RL studies.

In modern applications such as energy and autonomous control, $N$ tends to be extremely large or even infinity, which stimulates the interest of solving (1) in *real time*. Let $f_k(\boldsymbol{x}_k, \boldsymbol{u}_k) = f_k(\boldsymbol{x}_k, \boldsymbol{u}_k; \boldsymbol{d}_k)$ where $\boldsymbol{d}_k$ is data at stage $k$, then a more realistic setting is to collect $\boldsymbol{d}_k$ as a data stream. That is, instead of knowing $f_0$ all the way to $f_{N-1}$ and solving (1) offline, we only know $f_k$ sequentially and have to solve (1) online. Model predictive control (MPC), also known as receding horizon

35th Conference on Neural Information Processing Systems (NeurIPS 2021).

control, is an important feedback control technique that is used in online optimization of control problems. Given the current system state, MPC looks ahead multiple stages on the prediction horizon, solves the subproblem on the prediction horizon to ***optimality***, and provides the first optimal control action to the system as the feedback. Then the system evolves to a new state based on the control action and the procedure is repeated. However, the classical nonlinear MPC is being regarded as a theoretical concept rather than a practical strategy. It is not as widely used as linear MPC, and is mostly investigated in slow dynamics. This is due to the intensive computations of solving the control subproblems when we have nonlinear models. Fortunately, the design of real-time iteration (RTI) and its predecessors [26, 37] have enabled RTI-based nonlinear MPC to be applied on fast dynamics and achieve good performance [48, 4, 14, 38, 15, 24, 18, 22, 17, 20, 16, 19, 39, 1].

Instead of using optimal feedback control, RTI schemes compute a cheap approximation of control action and provide back to the system as fast as possible. The existing RTI schemes perform *a single, full Newton step* for each subproblem, and the output iterate of the previous subproblem is used to warmly initialize the current subproblem [8, 41, 42]. Thus, RTI schemes significantly reduce the computations of each subproblem, which makes them succeed in modern industrial applications. RTI schemes exploit the fact that two successive subproblems are closely related though different, so that the iteration can achieve the convergence "on the fly" (i.e. the error goes to zero as horizon moves forward). However, the RTI-based MPC is challenging to analyze. The main difficulty lies in the fact that the prediction horizon is shifted each time. Each subproblem is appended by a new stage that has never been scanned before and may introduce perturbation into the system. Different RTI-based MPC schemes have been studied [9, 12, 23], with an improved local analysis established in [31] recently.

Our paper complements the literature on RTI-based MPC by investigating its global convergence. We emphasize that by global convergence we mean *the convergence to stationary points (i.e. KKT points) for any initialization*, instead of the convergence to global optimality. The latter is not achievable for nonlinear problems without strong assumptions [34]. Instead of using a full Newton step, we design an adaptive scheme to involve the stepsize selection procedure. In fact, the stepsize selection has been suggested as a way to improve the global behavior of RTI in shrinking (not moving) horizons in an old paper [25]. The authors implemented a $\ell_1$ penalized merit function[1] with a watchdog line search. However, the global convergence theory remains open. Different from the suggestion of [25], we use a *differentiable exact augmented Lagrangian* as the merit function. This function mainly has two differences to the $\ell_1$ penalized merit function: (i) it depends on both primal and dual variables, so that dual variables are updated using the same stepsize as the primal variables ($\ell_1$ merit function only updates the primal variables); (ii) a simple backtracking line search can effectively overcome Maratos effect [29] and accept the unit stepsize locally ($\ell_1$ merit function requires a more time-consuming line search method such as watchdog or second-order correction). We conduct experiments to show the superiority of augmented Lagrangian over $\ell_1$ penalized function, especially for time-varying problems. Moreover, our algorithm adaptively selects the penalty parameters of augmented Lagrangian on the fly. When the unit stepsize is accepted, the algorithm simply boils down to the standard RTI-based MPC, and the existing local theories [8, 31] apply seamlessly. We notice that the same merit function has been employed in a recent work [32] to achieve the global convergence for solving (1) in offline fashion, with a horizon decomposition strategy. In that work, the authors showed the unit stepsize is accepted even with an inexact Newton direction. Our paper is in online regime and does not investigate the local behavior.

**Related work and contribution:** Our work is closely related to a growing literature on online optimization via RTI. The theories of RTI have been investigated over the past decade. [10] and [8] established the local convergence of RTI in shrinking horizons and moving horizons, respectively. When we have different models, [3, 42, 44, 35, 36] all showed similar results. A different view of RTI-based MPC is via the parametric optimization, where subproblems are parameterized by a continuous time index and a single Newton step is performed at each time instant. Similar stability results in this setup are shown in [11, 45, 13]. Recently, [31] studied a lag-$L$ RTI-based MPC (i.e. one shifts $L$ stages each time), and proved that RTI iterates on each stage converge linearly with a rate decaying exponentially in the prediction horizon length. Their analyses rely on the sensitivity analysis of (1) in [30, 33]. An important consequence of convergence of online iteration is the dynamic regret analysis [47, 27, 7, 43]. For example, the convergence rate in [31] and the regret order in [27] are both exponential in prediction horizon length. This is due to the widely-used Lipschitz continuity assumption on the objective.

---

[1]The merit function refers to the function that is used in the line search.

However, all above theories are local results. The algorithms always take a full (Newton) step for each subproblem, without involving any stepsize selection procedure. Thus, the analyses require to have an initial point that is sufficiently close to the local solution, or require that the first Newton step is sufficiently small. Obviously, for Problem (1), when $\|\bar{\boldsymbol{x}}_0\|$ is small enough, the origin is a good initial point and RTI iterates stably track the origin. However, if $\|\bar{\boldsymbol{x}}_0\|$ is large or we may want to track a nontrivial MPC policy, the origin is not a reasonable choice for the initial point. Based on these considerations, the global convergence of RTI addressed in this paper complements the aforementioned rich local analyses. Again, by global convergence we mean that the KKT residuals of subproblems go to zero as horizon moves forward for *any* initialization. The known, local stability of tracking error of RTI is not addressed in this paper, but referred to [45, Theorem 4.2].

The challenge of our analysis is that: *the problem varies with time instants and only a single Newton step is performed at each time instant.* Such a setup contrasts to online Newton (with line search) analysis, where a single Newton step targeting the *same* problem is performed. A key, technical step of our analysis is to show that the *exact* augmented Lagrangian merit function keeps decreasing as horizons shift. To our knowledge, this is the first paper studying the global behavior of online iteration under RTI setting, via a tool of the exact augmented Lagrangian. Another contribution to nonlinear MPC is that we consider time-varying dynamics, while most of the literature considered time-invariant dynamics (e.g. [8, 21, 40]). The subproblems of time-invariant dynamics only differ in the initial state, which results in a much simpler analysis. We notice that [31] allowed time-varying dynamics, but required to use a larger lag when shifting the horizon. We allow lag one instead.

**Notations:** For two vectors $\boldsymbol{a}, \boldsymbol{b}$, $(\boldsymbol{a}; \boldsymbol{b})$ denotes a long column vector that stacks $\boldsymbol{a}$ and $\boldsymbol{b}$ together. For two scalars $a, b$, $a \vee b = \max(a, b)$. Without specification, $\|\cdot\|$ denotes the $\ell_2$ norm for vectors and operator norm for matrices. We also reserve the following notations. $\boldsymbol{0}$ denotes zero vector or matrix, and $I$ denotes the identity matrix. Their dimensions are clear from the context. $M$ denotes the prediction horizon length, $t$ indexes the sampling time, and $\boldsymbol{z}_k = (\boldsymbol{x}_k; \boldsymbol{u}_k) \in \mathbb{R}^{n_z}$ with $n_z = n_x + n_u$ denotes the state-control pair at stage $k$.

## 2  Preliminaries

We introduce MPC subproblems formulation, the Newton step, and the augmented Lagrangian merit function in this section. At time $t$, the prediction horizon shifts to $[t, M_t]$ with $M_t = t + M$ denoting the last stage, and the corresponding subproblem $\mathcal{P}^t(\bar{\boldsymbol{x}}_t)$ is given by

$$\mathcal{P}^t(\bar{\boldsymbol{x}}_t): \quad \min_{\substack{\boldsymbol{x}_{t:M_t} \\ \boldsymbol{u}_{t:M_t-1}}} \quad \sum_{k=t}^{M_t-1} g_k(\boldsymbol{x}_k, \boldsymbol{u}_k) + g_{M_t}(\boldsymbol{x}_{M_t}, \boldsymbol{0}) + \frac{\mu}{2}\|\boldsymbol{x}_{M_t}\|^2,$$
$$\text{s.t.} \quad \boldsymbol{x}_{k+1} = f_k(\boldsymbol{x}_k, \boldsymbol{u}_k), \quad k \in [t, M_t - 1], \qquad (2)$$
$$\boldsymbol{x}_t = \bar{\boldsymbol{x}}_t.$$

Here, $\mu$ is a quadratic penalty parameter independent from $t$; $\bar{\boldsymbol{x}}_t$ is the current system state, which is given and determined by the control feedback provided to the system before $t$. The formulation (2) is used in [31] for local analysis, while alternative subproblem formulations are available as well [8, 23].

Given subproblem (2), RTI schemes perform a single Newton step for $\mathcal{P}^t(\bar{\boldsymbol{x}}_t)$ and move to the next time instant. To ease notations, we let $\tilde{\boldsymbol{x}}_t = \boldsymbol{x}_{t:M_t}$ and $\tilde{\boldsymbol{u}}_t = \boldsymbol{u}_{t:M_t-1}$ denote the state and control variables of the $t$-th subproblem, respectively. We also let $\tilde{\boldsymbol{z}}_t = \boldsymbol{z}_{t:M_t} = (\boldsymbol{z}_t; \ldots; \boldsymbol{z}_{M_t})$ denote the whole ordered state-control variables, with slightly abusing the notation $\boldsymbol{z}_{M_t}$ to let $\boldsymbol{z}_{M_t} = \boldsymbol{x}_{M_t}$. We sometimes also express $\tilde{\boldsymbol{z}}_t = (\tilde{\boldsymbol{x}}_t, \tilde{\boldsymbol{u}}_t)$ when we specify each component of $\tilde{\boldsymbol{z}}_t$. Then, the Lagrangian function of (2) is

$$\mathcal{L}^t(\tilde{\boldsymbol{z}}_t, \tilde{\boldsymbol{\lambda}}_t; \bar{\boldsymbol{x}}_t) = \sum_{k=t}^{M_t-1} g_k(\boldsymbol{x}_k, \boldsymbol{u}_k) + g_{M_t}(\boldsymbol{x}_{M_t}, \boldsymbol{0}) + \frac{\mu}{2}\|\boldsymbol{x}_{M_t}\|^2$$
$$+ \sum_{k=t}^{M_t-1} \boldsymbol{\lambda}_k^T(\boldsymbol{x}_{k+1} - f_k(\boldsymbol{x}_k, \boldsymbol{u}_k)) + \boldsymbol{\lambda}_{t-1}^T(\boldsymbol{x}_t - \bar{\boldsymbol{x}}_t),$$

where $\tilde{\boldsymbol{\lambda}}_t = \boldsymbol{\lambda}_{t-1:M_t-1}$ is the dual vector. Here, $\boldsymbol{\lambda}_k \in \mathbb{R}^{n_x}$ is associated to the $k$-th constraint of (2) for $k \in [t, M_t - 1]$, and $\boldsymbol{\lambda}_{t-1} \in \mathbb{R}^{n_x}$ is associated to the initial state constraint.

Given the input iterate $(\tilde{z}_t^0, \tilde{\lambda}_t^0)$, the Newton direction $(\Delta \tilde{z}_t, \Delta \tilde{\lambda}_t)$ is solved by the KKT system

$$\begin{pmatrix} B^t & (G^t)^T \\ G^t & \mathbf{0} \end{pmatrix} \begin{pmatrix} \Delta \tilde{z}_t \\ \Delta \tilde{\lambda}_t \end{pmatrix} = - \begin{pmatrix} \nabla_{\tilde{z}_t} \mathcal{L}^t \\ \nabla_{\tilde{\lambda}_t} \mathcal{L}^t \end{pmatrix}, \tag{3}$$

where $B^t$ is an approximation of the Hessian $H^t = \nabla^2_{\tilde{z}_t} \mathcal{L}^t$; $G^t = \nabla_{\tilde{\lambda}_t \tilde{z}_t} \mathcal{L}^t$ is the Jacobian matrix; and all matrices in (3) are evaluated at $(\tilde{z}_t^0, \tilde{\lambda}_t^0)$. The left hand side matrix is called the KKT matrix. In what follows, $H^t, B^t, G^t$ are always evaluated at the input iterate $(\tilde{z}_t^0, \tilde{\lambda}_t^0)$ if no evaluation point is specified. Taking a closer look at (3) (cf. Appendix A), we note that the KKT matrix and $\nabla_{\tilde{z}_t} \mathcal{L}^t$ are *independent* from the given initial state $\bar{x}_t$. This is a crucial observation. In reality, the realization of the system state may lag behind the sampling time, since the system has to apply the feedback control to evolve. Thus, $\bar{x}_t$ may not be known as soon as the input iterate $(\tilde{z}_t^0, \tilde{\lambda}_t^0)$. However, a promising property of RTI is that most of computations of (3) can be prepared without knowing $\bar{x}_t$.

Given the Newton direction $(\Delta \tilde{z}_t, \Delta \tilde{\lambda}_t)$ from (3), we update the iterate by involving a stepsize $\alpha_t$

$$\begin{pmatrix} \tilde{z}_t^1 \\ \tilde{\lambda}_t^1 \end{pmatrix} = \begin{pmatrix} \tilde{z}_t^0 \\ \tilde{\lambda}_t^0 \end{pmatrix} + \alpha_t \begin{pmatrix} \Delta \tilde{z}_t \\ \Delta \tilde{\lambda}_t \end{pmatrix}. \tag{4}$$

In our paper, $\alpha_t$ is selected by forcing *Armijo condition* hold on a differentiable exact augmented Lagrangian, defined as

$$\mathcal{L}_\eta^t(\tilde{z}_t, \tilde{\lambda}_t; \bar{x}_t) = \mathcal{L}^t(\tilde{z}_t, \tilde{\lambda}_t; \bar{x}_t) + \frac{\eta_1}{2} \left\| \nabla_{\tilde{\lambda}_t} \mathcal{L}^t(\tilde{z}_t, \tilde{\lambda}_t; \bar{x}_t) \right\|^2 + \frac{\eta_2}{2} \left\| \nabla_{\tilde{z}_t} \mathcal{L}^t(\tilde{z}_t, \tilde{\lambda}_t; \bar{x}_t) \right\|^2. \tag{5}$$

Here $\eta = (\eta_1, \eta_2)$ are parameters of the quadratic penalties. (5) is called *exact* augmented Lagrangian since one can show that, for sufficiently large $\eta_1$ and sufficiently small $\eta_2$, a strict local minimizer of $\mathcal{L}_\eta^t(\cdot)$ is also a strict local minimizer of Problem (2) and vice versa (see [2, Proposition 4.5]). When $\eta_2 = 0$, $\mathcal{L}_\eta^t(\cdot)$ reduces to standard augmented Lagrangian, while it does not enjoy exactness property anymore, and may cause issues when it is used as merit function. For simplicity, we sometimes suppress "exact" and call $\mathcal{L}_\eta^t(\cdot)$ augmented Lagrangian. Based on (5), $\alpha_t$ is selected by making the following Armijo condition,

$$\mathcal{L}_\eta^t(\tilde{z}_t^1, \tilde{\lambda}_t^1; \bar{x}_t) \leq \mathcal{L}_\eta^t(\tilde{z}_t^0, \tilde{\lambda}_t^0; \bar{x}_t) + \alpha_t \beta \begin{pmatrix} \nabla_{\tilde{z}_t} \mathcal{L}_\eta^t(\tilde{z}_t^0, \tilde{\lambda}_t^0; \bar{x}_t) \\ \nabla_{\tilde{\lambda}_t} \mathcal{L}_\eta^t(\tilde{z}_t^0, \tilde{\lambda}_t^0; \bar{x}_t) \end{pmatrix}^T \begin{pmatrix} \Delta \tilde{z}_t \\ \Delta \tilde{\lambda}_t \end{pmatrix}, \tag{6}$$

hold for a prespecified parameter $\beta \in (0, 1)$. Note that, by mean value theory, $\alpha_t$ can always be found by backtracking line search provided the inner product in (6) is negative (cf. Theorem 4.5).

## 3   An Adaptive RTI-based MPC Scheme

We now set the stage to propose our algorithm. The algorithm inherits the flavor of RTI schemes: it performs a single Newton step for each subproblem. Differently, it involves a stepsize selection procedure based on $\mathcal{L}_\eta^t(\cdot)$ instead of using a full step (i.e. $\alpha_t = 1$).

The algorithm adaptively selects the parameter $\eta$ on the fly (i.e. it automatically selects suitable $\eta$ for each subproblem). However, since $\mu$ is required in the problem formulation, the adaptivity on $\mu$ may require to solve the KKT system multiple times for each subproblem. Thus, the gain from the adaptivity on $\mu$ for our procedure is limited. We hence prefer to tune $\mu$ manually in practice. Our experiments show that our algorithm is robust in $\mu$ (see Figure 1).

Let us now illustrate how we transit from the triple $(\tilde{z}_t^0, \tilde{\lambda}_t^0; \bar{x}_t)$ to $(\tilde{z}_{t+1}^0, \tilde{\lambda}_{t+1}^0; \bar{x}_{t+1})$. To further ease notations, we let $\mathcal{L}_\eta^{t,1}, \mathcal{L}_\eta^{t,0}, \mathcal{L}^{t,1}, \mathcal{L}^{t,0}$ denote the function values of $\mathcal{L}_\eta^t(\cdot)$ and $\mathcal{L}^t(\cdot)$ at the output iterate $(\tilde{z}_t^1, \tilde{\lambda}_t^1)$ and the input iterate $(\tilde{z}_t^0, \tilde{\lambda}_t^0)$, respectively. In addition, when we specify the iterate on each stage, we let $\tilde{x}_t^{\text{Id}} = x_{t:M_t,t}^{\text{Id}}$ for Id = 0 or 1 (similar for $\tilde{z}_t^{\text{Id}}, \tilde{\lambda}_t^{\text{Id}}, \Delta \tilde{z}_t$, and $\Delta \tilde{\lambda}_t$). For example, $x_{t+1,t}^1$ denotes the output iterate of the $t$-th subproblem at stage $t + 1$.

Given the input iterate $(\tilde{z}_t^0, \tilde{\lambda}_t^0)$, we perform four steps:
**(1)** We compute $B^t, G^t, \nabla_{\tilde{z}_t} \mathcal{L}^{t,0}$, and $\nabla_{\tilde{\lambda}_t} \mathcal{L}^{t,0}$ except the first row. After we observe $\bar{x}_t$, we compute

---

**Algorithm 1** An Adaptive RTI-based MPC Scheme

---

1: **Input:** initial iterate $(\tilde{z}_0^0, \tilde{\lambda}_0^0)$, initial state $\bar{x}_0$, and parameters $\mu > 0$, $\rho > 1$, $\eta^0 = (\eta_1^0, \eta_2^0) = (\mu^2, 1)$, and $\beta \in (0, 1)$;

2: **for** $t = 0, 1, 2, \ldots$ **do**

3:     Compute $B^t$, $G^t$, $\nabla_{\tilde{z}_t}\mathcal{L}^{t,0}$, $\nabla_{\tilde{\lambda}_t}\mathcal{L}^{t,0}$, and solve the KKT system (3) to obtain $(\Delta\tilde{z}_t, \Delta\tilde{\lambda}_t)$;

4:     **while** $\begin{pmatrix} \nabla_{\tilde{z}_t}\mathcal{L}_{\eta^t}^{t,0} \\ \nabla_{\tilde{\lambda}_t}\mathcal{L}_{\eta^t}^{t,0} \end{pmatrix}^T \begin{pmatrix} \Delta\tilde{z}_t \\ \Delta\tilde{\lambda}_t \end{pmatrix} > -\frac{\eta_2^t}{4} \left\| \begin{pmatrix} \nabla_{\tilde{z}_t}\mathcal{L}^{t,0} \\ \nabla_{\tilde{\lambda}_t}\mathcal{L}^{t,0} \end{pmatrix} \right\|^2$ **do**

5:         Let $\eta_1^t = \rho^2\eta_1^t$, $\eta_2^t = \eta_2^t/\rho$;

6:     **end while**

7:     Select $\alpha_t$ by backtracking line search to make (6) hold, and update iterates by (4);

8:     Define $(\tilde{z}_{t+1}^0, \tilde{\lambda}_{t+1}^0)$ and $\bar{x}_{t+1}$ according to (8), and let $\eta^{t+1} = \eta^t$;

9: **end for**

---

the first row of $\nabla_{\tilde{\lambda}_t}\mathcal{L}^{t,0}$. Then, we compute the Newton direction $(\Delta\tilde{z}_t, \Delta\tilde{\lambda}_t)$ by solving (3).

**(2)** We select $\eta^t = (\eta_1^t, \eta_2^t)$ such that

$$\begin{pmatrix} \nabla_{\tilde{z}_t}\mathcal{L}_{\eta^t}^{t,0} \\ \nabla_{\tilde{\lambda}_t}\mathcal{L}_{\eta^t}^{t,0} \end{pmatrix}^T \begin{pmatrix} \Delta\tilde{z}_t \\ \Delta\tilde{\lambda}_t \end{pmatrix} \leq -\frac{\eta_2^t}{4} \left\| \begin{pmatrix} \nabla_{\tilde{z}_t}\mathcal{L}^{t,0} \\ \nabla_{\tilde{\lambda}_t}\mathcal{L}^{t,0} \end{pmatrix} \right\|^2. \tag{7}$$

As shown in Algorithm 1, we can use a While loop to increase $\eta_1^t$ and decrease $\eta_2^t$ until (7) is satisfied.
**(3)** Using $\eta^t$ from the last step, we select $\alpha_t$ by backtracking line search to make Armijo condition (6) hold. We then obtain the output iterate $(\tilde{z}_t^1, \tilde{\lambda}_t^1)$ by (4).
**(4)** We transit to the next horizon by letting

$$\begin{aligned} \tilde{x}_{t+1}^0 &= x_{t+1:M_{t+1},t+1}^0 := (x_{t+1,t}^1; x_{t+2,t}^1; \ldots; x_{M_t,t}^1; \mathbf{0}), \\ \tilde{u}_{t+1}^0 &= u_{t+1:M_{t+1}-1,t+1}^0 := (u_{t+1,t}^1; u_{t+2,t}^1; \ldots; u_{M_t-1,t}^1; \mathbf{0}), \\ \tilde{\lambda}_{t+1}^0 &= \lambda_{t:M_{t+1}-1,t+1}^0 := (\lambda_{t,t}^1; \lambda_{t+1,t}^1; \ldots; \lambda_{M_t-1,t}^1; \mathbf{0}), \\ \bar{x}_{t+1} &:= f_t(x_{t,t}^1, u_{t,t}^1). \end{aligned} \tag{8}$$

The above four steps are displayed in Algorithm 1. If we remove Step (2) and use $\alpha_t = 1$ in Step (3), it reduces to standard RTI schemes [8, 41, 42, 31]. However, as shown later, the difference that we select $\alpha_t$ based on augmented Lagrangian $\mathcal{L}_{\eta^t}^t(\cdot)$ enables Algorithm 1 to converge globally.

**Remark 3.1.** We discuss the overhead of the algorithm. For each subproblem, the computational complexity is dominated by solving the KKT system. The KKT matrix is a square matrix with dimensions $2(M + 1)n_x + Mn_u$. Directly inverting the matrix results in $O((M(n_x + n_u))^3)$ flops. Fortunately, we can exploit the sparse structure of the KKT matrix and apply sparse $LDL^T$ factorization as did in Section C of [41], which results in $O(M(n_x + n_u)^3)$ flops instead. So the computations are linear in terms of the prediction horizon length $M$.

## 4 Global Convergence Analysis

We conduct global convergence analysis for Algorithm 1. We show that, for *any* initial iterate $(\tilde{z}_0^0, \tilde{\lambda}_0^0)$, the KKT residual of the $t$-th subproblem, $\|\nabla\mathcal{L}^{t,0}\|$, converges to zero as $t$ increases. The significance of the result is that we only perform a single Newton step for each time-varying subproblem, and our result complements the local results of RTI. We should mention that, like most of deterministic methods for nonlinear problems [34], Algorithm 1 is not selective to stationary points, and may converge to saddle points. Converging to local minimum is only possible under stronger, local assumptions, which are not addressed in this paper but we refer to [31].

### 4.1 Assumptions

We require the following assumptions.

**Assumption 4.1** (Compactness). We assume all iterates lie in a convex compact set that contains the origin. In particular, we assume there exist convex compact sets $\mathbf{0} \in \mathcal{Z} \subseteq \mathbb{R}^{n_x + n_u}$ and $\mathbf{0} \in \Lambda \subseteq \mathbb{R}^{n_x}$, such that $(\boldsymbol{z}_{k,t}^0, \boldsymbol{\lambda}_{k,t}^0) \in \mathcal{Z} \times \Lambda$ for any $t \geq 0$ and any stage $k$ of the $t$-th subproblem.

**Assumption 4.2** (Boundedness condition). We assume there exist absolute constants $\Upsilon, \delta, \gamma_{RH} > 0$ such that (i) for any $t \geq 0$, $B^t$ satisfies $\boldsymbol{z}^T B^t \boldsymbol{z} \geq \gamma_{RH} \|\boldsymbol{z}\|^2$, $\forall \boldsymbol{z} \in \{\boldsymbol{z} : G^t \boldsymbol{z} = \mathbf{0}, \boldsymbol{z} \neq \mathbf{0}\}$, and $\|B^t - H^t\| \leq \delta$; (ii) for any $k \geq 0$ and $\boldsymbol{z}_k \in \mathcal{Z}$, we have

$$\|\nabla_{\boldsymbol{x}_k} f_k(\boldsymbol{z}_k)\| \vee \|\nabla_{\boldsymbol{u}_k} f_k(\boldsymbol{z}_k)\| \vee \|\nabla_{\boldsymbol{z}_k}^2 g_k(\boldsymbol{z}_k)\| \vee \{\|\nabla_{\boldsymbol{z}_k}^2 f_{k,l}(\boldsymbol{z}_k)\|\}_{l=1}^{n_x} \leq \Upsilon, \tag{9}$$

$$\{\|\nabla_{\boldsymbol{z}_k}^2 [\nabla_{\boldsymbol{z}_k} g_k(\boldsymbol{z}_k)]_i\|, \|\nabla_{\boldsymbol{z}_k}^2 [\nabla_{\boldsymbol{z}_k} f_{k,l}(\boldsymbol{z}_k)]_i\|\}_{i=1,l=1}^{n_z, n_x} \leq \Upsilon; \tag{10}$$

(iii) for any $t \geq 0$, $G^t (G^t)^T \succeq I/\Upsilon^2$.

**Assumption 4.3.** We assume there exist absolute constants $c, C > 0$ such that for any $t \geq 0$ (i) $\|\Delta \tilde{\boldsymbol{\lambda}}_{M_t-1,t}\| \leq c\|(\Delta \tilde{\boldsymbol{z}}_t; \Delta \tilde{\boldsymbol{\lambda}}_t)\|$ if $\boldsymbol{x}_{M_t,t}^1 \neq \mathbf{0}$. (ii) $\|\boldsymbol{\lambda}_{t-1,t}^1\| \leq C\|\boldsymbol{x}_{t,t}^1 - \bar{\boldsymbol{x}}_t\|$ if $\boldsymbol{x}_{t,t}^1 \neq \bar{\boldsymbol{x}}_t$.

A 4.1 and 4.2 are standard in the literature on Newton analysis [34, 2]. We note that Jacobian $G^t$ always has full row rank (see (13) in Appendix A), so the linear independence constraint qualification (LICQ) holds. Together with LICQ, A 4.2(i) implies the KKT matrix is nonsingular [34, Lemma 16.1]. In fact, we can simply let $B^t = \mu I$, which satisfies all conditions trivially; is sufficient for our global analysis; and is indeed used in our experiments. More generally, $B^t$ can approximate the Hessian $H^t$ using any methods in [34]. Different from existing analyses which only require the bounded Hessian, (10) requires one more derivative—the third derivative—to be bounded. This is standard when using augmented Lagrangian $\mathcal{L}_\eta^t(\cdot)$ in the algorithm [46, 2, 32], because the Hessian of $\mathcal{L}_\eta^t(\cdot)$ requires the third derivatives of $g_k, f_k$. We mention that the third derivatives are only used in the analysis, and do not have to be computed in the implementation. (9) is directly implied by the compactness in A 4.1. A 4.2(iii) is also reasonable to hold since $G^t$ has full row rank. For time-invariant problems where $g_k = g$, $f_k = f$, A 4.2(ii) and (iii) hold trivially.

A 4.3 is particularly imposed for MPC analysis; a condition with the same flavor is in [8]. At the first glance, it seems to be a restrictive assumption. Fortunately, it is guaranteed to hold when $\mu$ is sufficiently large (the threshold depends on problem parameters), and is easily checkable during the iteration. In particular, A 4.3(i) assumes that the last dual step $\Delta \tilde{\boldsymbol{\lambda}}_{M_t-1,t}$ is bounded by the full Newton step $(\Delta \tilde{\boldsymbol{z}}_t, \Delta \tilde{\boldsymbol{\lambda}}_t)$. It holds trivially for $c = 1$, however, we hope $c$ is small. One can expect this condition to be satisfied for large enough $\mu$, since $\Delta \tilde{\boldsymbol{\lambda}}_{M_t-1,t}$ measures the disturbance of $\boldsymbol{x}_{M_t,t}^1$ away from $\boldsymbol{x}_{M_t,t}^0 = \mathbf{0}$. When $\mu$ is large, we know from the quadratic penalty in (2) that $\boldsymbol{x}_{M_t,t}^1 \approx \mathbf{0}$. Hence, $\Delta \tilde{\boldsymbol{\lambda}}_{M_t-1,t}$ becomes arbitrarily small. See more discussions in [8, Assumption 3] for local analysis of MPC. A 4.3(ii) allows any constant $C$ as long as $C \leq \mu^2$, so it holds as well for large enough $\mu$. In addition, A 4.3(ii) is not required if $\alpha_t = 1$ is selected, which is always true when $(\tilde{\boldsymbol{z}}_t^0, \tilde{\boldsymbol{\lambda}}_t^0)$ is close to a stationary point of the subproblem[2]. This is because that, if $\alpha_t = 1$, then $\boldsymbol{x}_{t,t}^1 = \bar{\boldsymbol{x}}_t$ for any input iterate $\boldsymbol{x}_{t,t}^0$.

## 4.2 Global convergence

We first show that Algorithm 1 is well-posed, that is, each iteration can be performed successfully. The computation of the Newton direction in Line 3 is ensured by the nonsingularity of the KKT matrix, implied by A 4.2 as discussed before. The backtracking line search in Line 7 is guaranteed by the mean value theorem [5]. It hence suffices to show that the While loop in Line 4 terminates in finite time.

**Theorem 4.4.** Consider Algorithm 1 under Assumptions 4.1 and 4.2. Let us define thresholds $\tau_1, \tau_2, \tau_3 > 0$ by

$$\tau_1 = \frac{25 \mu^2 \Upsilon^2}{\gamma_{RH}}, \quad \tau_2 = \frac{\gamma_{RH}}{2\delta^2}, \quad \tau_3 = \frac{32 \mu^2 \Upsilon^2}{\gamma_{RH}^2}.$$

For any $t \geq 0$, if $\eta_1 \geq \tau_1$, $\eta_2 \leq \tau_2$, $\eta_1 \eta_2 \geq \tau_3$, then

$$\begin{pmatrix} \nabla_{\tilde{\boldsymbol{z}}_t} \mathcal{L}_\eta^{t,0} \\ \nabla_{\tilde{\boldsymbol{\lambda}}_t} \mathcal{L}_\eta^{t,0} \end{pmatrix}^T \begin{pmatrix} \Delta \tilde{\boldsymbol{z}}_t \\ \Delta \tilde{\boldsymbol{\lambda}}_t \end{pmatrix} \leq -\frac{\eta_2}{4} \left\| \begin{pmatrix} \nabla_{\tilde{\boldsymbol{z}}_t} \mathcal{L}^{t,0} \\ \nabla_{\tilde{\boldsymbol{\lambda}}_t} \mathcal{L}^{t,0} \end{pmatrix} \right\|^2.$$

---

[2][28] showed under certain conditions that the backtracking line search would always select a unit stepsize locally when using exact augmented Lagrangian as the merit function.

Further, there exist an integer $\tau > 0$ and positive constants $\bar{\eta} = (\bar{\eta}_1, \bar{\eta}_2)$ satisfying

$$\bar{\eta}_1 \leq \frac{32^2 \rho^2 \mu^2 \Upsilon^4}{\gamma_{RH}^4}, \quad \bar{\eta}_2 \geq \frac{\gamma_{RH}}{32 \rho \Upsilon^2},$$

such that $\eta^t = \bar{\eta}, \forall t \geq \tau$.

Theorem 4.4 shows that the While loop indeed terminates in finite time, and more importantly, the penalty parameter $\eta^t$ stabilizes after certain iterations. This result is critical for global convergence since, after $\tau$ subproblems, all the succeeding subproblems try to decrease a much similar merit function. We note that $\bar{\eta}_1$ depends on $\mu$, so the penalty of the objective connects to the penalty of augmented Lagrangian. We mention that the constants in $\{\tau_i\}_{i=1}^3$ and $\bar{\eta}_1, \bar{\eta}_2$ can be significantly improved by a careful calculation. However, we leave it for future work since it's not main target here.

It is also worth mentioning that the stabilized parameters $(\bar{\eta}_1, \bar{\eta}_2)$ may not ensure the equivalence between stationary points of $\mathcal{L}_{\bar{\eta}}^t(\cdot)$ and KKT points of $\mathcal{P}^t(\bar{x}_t)$, because the stabilized values rely on particular iteration sequence. The parameters thresholds for the exactness property of $\mathcal{L}_\eta^t$ ensure that any stationary points of augmented Lagrangian are equivalent to KKT points, which is a stronger guarantee. However, the stabilized parameters indeed ensure that our particular iteration converges to a KKT point, not just a stationary point of augmented Lagrangian.

We now study how the merit function decreases from one subproblem to another.

**Theorem 4.5.** Consider Algorithm 1 under Assumptions 4.1 and 4.2. There exists a constant $\bar{\alpha} > 0$ such that $\alpha_t \geq \bar{\alpha}, \forall t \geq \tau$ where $\tau$ is from Theorem 4.4. Moreover,

$$\mathcal{L}_{\bar{\eta}}^{t,1} \leq \mathcal{L}_{\bar{\eta}}^{t,0} - \frac{\bar{\eta}_2 \bar{\alpha} \beta}{4} \|\nabla \mathcal{L}^{t,0}\|^2.$$

Theorem 4.5 shows that Armijo condition is satisfied for sufficiently small $\alpha_t$, with a threshold independent of $t$. Moreover, after one Newton step, we can observe a sufficient decrease of the merit function. However, since we only perform one Newton step and transit to the next subproblem immediately, we cannot claim the decrease is sustained. This is indeed the main challenge of global analysis of RTI schemes. Different subproblems target different merit functions, even $\eta$ is shared.

We show in the next lemma that, the transition between subproblems may increase the merit function, but not too much. Overall, the merit function $\mathcal{L}_{\bar{\eta}}^{t,0}$ keeps decreasing.

**Lemma 4.6.** Under Assumptions 4.1-4.3 and let $\kappa = \frac{\mu}{\gamma_{RH}}$ denote the condition number of the Hessian matrix. Suppose the constants $c, C$ in Assumption 4.3 satisfies $c \lesssim \gamma_{RH}/\kappa^3$ and $C \lesssim \mu^2$. Then,[3]

$$\mathcal{L}_{\bar{\eta}}^{t+1,0} \leq \mathcal{L}_{\bar{\eta}}^{t,1} + \frac{\bar{\eta}_2 \bar{\alpha} \beta}{8} \|\nabla \mathcal{L}^{t,0}\|^2.$$

Combining Lemma 4.6 with Theorem 4.5, we immediately have

$$\text{one-step error recursion:} \quad \mathcal{L}_{\bar{\eta}}^{t+1,0} \leq \mathcal{L}_{\bar{\eta}}^{t,0} - \frac{\bar{\eta}_2 \bar{\alpha} \beta}{8} \|\nabla \mathcal{L}^{t,0}\|^2. \tag{11}$$

Using (11), we sum over $t$ and finally establish the global convergence in the next theorem.

**Theorem 4.7.** Consider Algorithm 1 under Assumptions 4.1-4.3. Suppose $c \lesssim \gamma_{RH}/\kappa^3$ and $C \lesssim \mu^2$, then for any input iterate $(\tilde{z}_0^0, \tilde{\lambda}_0^0)$, we have $\|\nabla \mathcal{L}^{t,0}\| \to 0$ as $t \to \infty$.

## 5 Numerical Experiment

We apply Algorithm 1 on a dynamic program to demonstrate the global behavior of RTI-based MPC. We compare augmented Lagrangian merit function with $\ell_1$ penalized merit function, defined as

$$\bar{\mathcal{L}}_\nu^t(\tilde{z}_t; \bar{x}_t) = \sum_{k=t}^{M_t-1} g_k(x_k, u_k) + g_{M_t}(x_{M_t}, \mathbf{0}) + \frac{\mu}{2} \|x_{M_t}\|^2$$

$$+ \nu \left( \sum_{k=t}^{M_t-1} \|x_{k+1} - f_k(x_k, u_k)\|_1 + \|x_t - \bar{x}_t\|_1 \right).$$

---

[3]"$\lesssim$" means the inequality holds up to a problem-independent constant.

Table 1: Simulation Setups

| CASE | $N$ | $M$ | $(a_k, b_k, c_k)$ | $\mu$ | $\eta_1$ | $\eta_2$ | $\beta$ | $\nu^0$ | $\rho$ | $\bar{x}_0$ | $\epsilon$ |
|------|-----|-----|-------------------|-------|----------|----------|---------|---------|--------|-------------|------------|
| CASE 1 | 100 | $(5,10,15)$ | $(1,1,1)$ | 5 | 25 | 1 | 0.4 | 1 | 1.5 | 10 | $10^{-8}$ |
| CASE 2 | 250 | $(5,10,15)$ | $(k,k,k)$ | 1 | 1 | 1 | 0.4 | 1 | 1.5 | 10 | $10^{-8}$ |
| CASE 3 | 250 | $(5,10,15)$ | $(k^2,k^2,k^2)$ | 20 | 100 | 1 | 0.4 | 1 | 1.5 | 10 | $10^{-8}$ |

Using $\bar{\mathcal{L}}_\nu^t(\cdot)$ as the merit function, Armijo condition is rewritten as $\bar{\mathcal{L}}_\nu^{t,1} \leq \bar{\mathcal{L}}_\nu^{t,0} + \alpha_t\beta \cdot D(\bar{\mathcal{L}}_\nu^{t,0}, \Delta\tilde{z}_t)$, where $D(\bar{\mathcal{L}}_\nu^{t,0}, \Delta\tilde{z}_t)$ is the directional derivative of $\bar{\mathcal{L}}_\nu^t(\cdot)$ at $\tilde{z}_t^0$ along the direction $\Delta\tilde{z}_t$, given by

$$D(\bar{\mathcal{L}}_\nu^{t,0}, \Delta\tilde{z}_t) = \sum_{k=t}^{M_t-1} \nabla^T g_k(z_{k,t}^0)\Delta z_{k,t} + \nabla_{\boldsymbol{x}_{M_t}}^T g_{M_t}(\boldsymbol{x}_{M_t,t}^0, \boldsymbol{0})\Delta\boldsymbol{x}_{M_t,t}$$

$$- \nu\left(\sum_{k=t}^{M_t-1} \|\boldsymbol{x}_{k+1} - f_k(\boldsymbol{x}_k, \boldsymbol{u}_k)\|_1 + \|\boldsymbol{x}_t - \bar{\boldsymbol{x}}_t\|_1\right) =: D_1 - \nu D_2.$$

To ensure $D(\bar{\mathcal{L}}_\nu^{t,0}, \Delta\tilde{z}_t)$ to be negative, we let $\nu^t = \max(D_1/\{(\rho-1)D_2\}, \nu^{t-1})$ with $\rho \in (1,2)$. Thus, RTI-based MPC using $\ell_1$ penalized merit function (called $\ell_1$ MPC) is obtained by replacing Lines 4 and 7 of Algorithm 1 by the above corresponding steps.

**Simulation setups.** We consider 1D trigonometric perturbed LQR problem. In particular, we let $g_k(z_k) = a_k x_k^2 + b_k u_k^2 + c_k \sin^2(x_k)$ and $f_k = x_k + u_k + \sin(x_k)$. We study three cases summarized in Table 1. For each case, we perform 1000 independent runs with randomly generalized initial iterate $(\tilde{z}_0^0, \tilde{\lambda}_0^0)$, by letting $(\boldsymbol{x}_{k,0}^0, \boldsymbol{u}_{k,0}^0, \boldsymbol{\lambda}_{k,0}^0) \sim \mathcal{N}(0, 25I), \forall k$. We stop the iteration if either $t > N - M$ (i.e. attains the iteration threshold) or $\|\nabla\mathcal{L}^{t,0}\| \leq \epsilon = 10^{-8}$ (i.e. attains the error threshold). We let $B^t = \mu I, \forall t$. For each case, we try three prediction horizon lengths, $M = 5, 10, 15$. We also vary $\mu$ in different cases to test the robustness of Algorithm 1 on $\mu$. For Case 1, we let $a_k = b_k = c_k = 1$, which results in a time-invariant problem. For Cases 2 and 3, we let $a_k = b_k = c_k = k$ and $k^2$, which results in time-varying problems. The linear and quadratic increase of coefficients bring significant challenges for solving subproblems online. The code is implemented in Julia 1.5.4 and is publicly available (with high resolution figures) at `https://github.com/senna1128/Global-RTI-MPC`.

**Evaluation of results.** For each case, we compare the average running time between two merit functions, and compare the maximum (over 1000 independent runs) of the number of iterations. The comparisons are shown in Table 2. From Table 2, we see that two merit functions work equally well for the time-invariant problem in Case 1, while augmented Lagrangian works significantly better than $\ell_1$ MPC for the time-varying problems in Cases 2 and 3. The latter fails to attain $10^{-8}$ KKT residuals within 250 iterations in both Case 2 and Case 3.

Moreover, we plot $\|\nabla\mathcal{L}^{t,0}\|$ v.s. $t$ in log scale for all cases. To have a better visualization, among 1000 initializations, we only plot the first 5 in Figure 1, but draw an error bar plot for all initializations in Figure 2. We see from Figure 1 (a)-(f) and Figure 2 (a)-(f) that both MPC schemes achieve global convergence within 25 iterations for the time-invariant problem in Case 1. Their (linear) convergence speed is comparable as well. However, for the time-varying problems in Cases 2 and 3, Figure 1 (g)-(r) and Figure 2 (g)-(r) suggest that augmented Lagrangian MPC performs much better than $\ell_1$ MPC. Although both schemes decrease KKT residuals as $t$ increases, which validates our global convergence theory, augmented Lagrangian MPC decreases the residuals linearly in $t$, and below the threshold $\epsilon = 10^{-8}$ within a moderate number of iterations. As a comparison, $\ell_1$ MPC decreases the residuals only to $[10^{-2}, 10^{-4}]$ using 250 iterations. Moreover, it is clear to see that $\ell_1$ MPC has a slow convergence speed locally (because the tail of $\|\nabla\mathcal{L}^{t,0}\|$ is flat), which does not appear in augmented Lagrangian MPC. Therefore, for time-varying problems, the proposed augmented Lagrangian MPC is much preferable than $\ell_1$ MPC.

We also take Case 2 as an example to show our algorithm is robust to $\mu$. We vary $\mu = 1, 5, 10, 100, 500, 1000$ and randomly generate an initial point for each $\mu$. We see from Figure 1 (s)-(x) that the performance of our algorithm is stable for different choices of $\mu$.

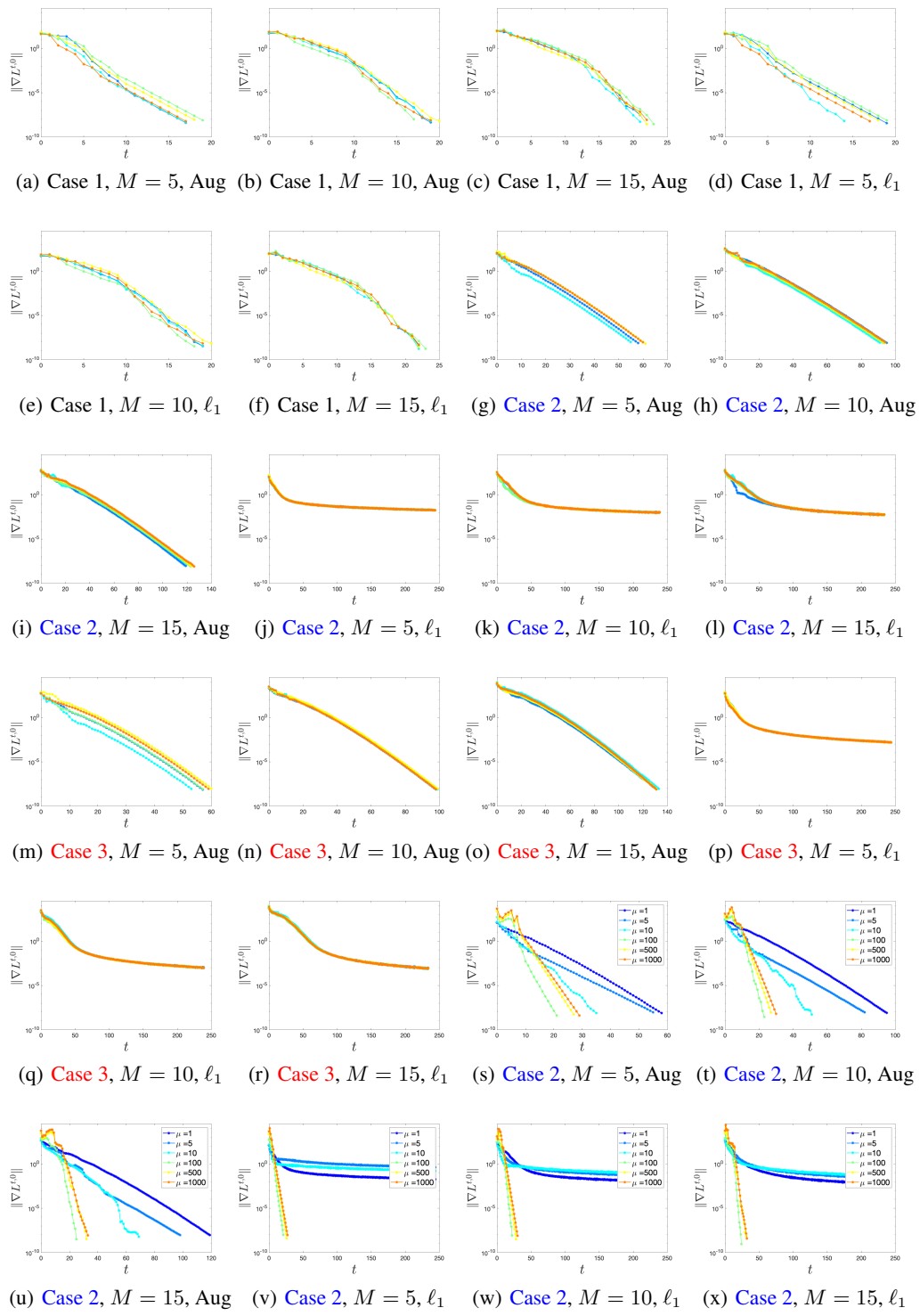

Figure 1: KKT residual plot. Each panel corresponds to a combination of Case, $M$ and the merit function. The figures (a)-(r) correspond to the settings in Table 1. In each panel, the five lines correspond to five different initial points that are randomly generalized. The figures (s)-(x) tests the robustness of $\mu$ using Case 2. In each panel, the five lines correspond to five $\mu$ varying from 1 to 5, with randomly generated initial point.

Table 2: Simulation Results

| $M$ | Case | log(Time) Aug | log(Time) $\ell_1$ | # of Iter Aug | # of Iter $\ell_1$ | Case | log(Time) Aug | log(Time) $\ell_1$ | # of Iter Aug | # of Iter $\ell_1$ | Case | log(Time) Aug | log(Time) $\ell_1$ | # of Iter Aug | # of Iter $\ell_1$ |
|---|---|---|---|---|---|---|---|---|---|---|---|---|---|---|---|
| 5 |  | -7 | 7 | 21 | 22 |  | -5 | -4 | 80 | **246** |  | -5 | -4 | 77 | **246** |
| 10 | Case 1 | -7 | -7 | 22 | 22 | Case 2 | -4 | -4 | 157 | **241** | Case 3 | -4 | -4 | 156 | **241** |
| 15 |  | -6 | -6 | 25 | 25 |  | -4 | -4 | 164 | **236** |  | -3 | -3 | 217 | **236** |

(the boldface number indicates the iteration stops by attaining the iteration threshold)

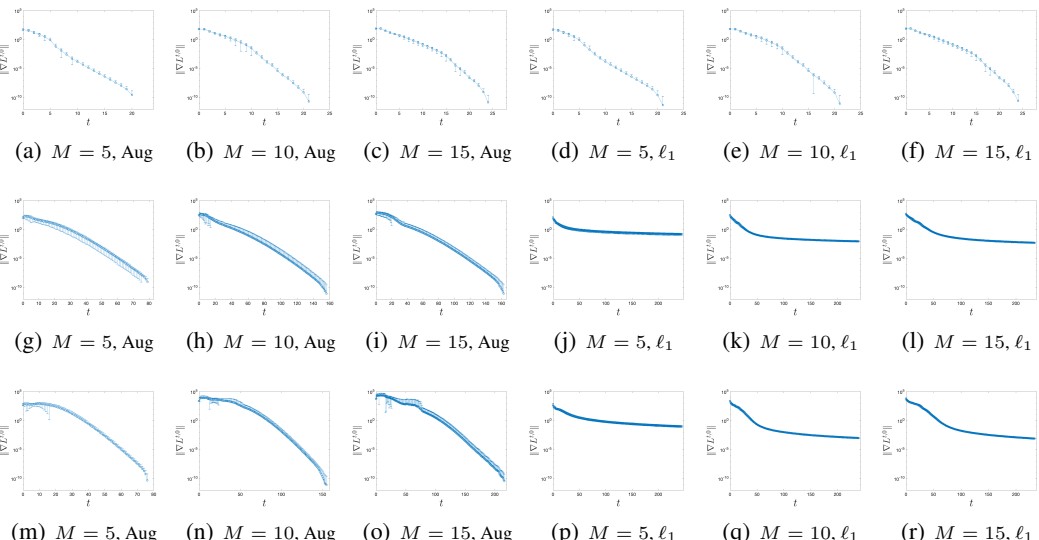

(a) $M = 5$, Aug  (b) $M = 10$, Aug  (c) $M = 15$, Aug  (d) $M = 5$, $\ell_1$  (e) $M = 10$, $\ell_1$  (f) $M = 15$, $\ell_1$

(g) $M = 5$, Aug  (h) $M = 10$, Aug  (i) $M = 15$, Aug  (j) $M = 5$, $\ell_1$  (k) $M = 10$, $\ell_1$  (l) $M = 15$, $\ell_1$

(m) $M = 5$, Aug  (n) $M = 10$, Aug  (o) $M = 15$, Aug  (p) $M = 5$, $\ell_1$  (q) $M = 10$, $\ell_1$  (r) $M = 15$, $\ell_1$

Figure 2: KKT residual bar plot. Each panel corresponds to a combination of Case, $M$ and the merit function. The first, second, third lines correspond to Case 1,2,3, respectively. In each panel, the circle corresponds to the mean among 1000 initializations, while the length of the vertical line of each point corresponds to the standard deviation. The low bound is suppressed if it is negative.

## 6   Conclusion

We studied the global behavior of RTI-based nonlinear MPC algorithm. Different from the existing schemes, we imposed line search for each subproblem to select the stepsize, using a differentiable exact augmented Lagrangian as the merit function. We proved that the proposed algorithm enjoys global convergence: for any initial iterate, the KKT residuals of subproblems converge to zero. Our analysis complements the existing local analyses of RTI-based MPC, which require a sufficiently good initial point.

One extension is to replace the Newton step by the quasi-Newton step. This may lead to a slow local convergence, however, the global convergence should be preserved. Another important extension is to study stochastic dynamics, like [40]. We can evaluate the iteration by either regret [27] or $\mathbb{E}\|\nabla\mathcal{L}^{t,0}\|$. Then even the local stability property of RTI-based MPC is not well understood. Furthermore, studying RTI schemes for model-based RL is an attractive research direction and, to some extent, our analysis is still applicable. However, how to apply RTI on model-free RL is still not clear to our knowledge. Finally, our experiments show that RTI-based MPC converge globally even for moderate $\mu$ (e.g. $\mu = 1$), which in some sense suggests Assumption 4.3 is not necessary. How to remove Assumption 4.3 is worth investigating.

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
