# Supplementary material:
## Global Convergence of Online Optimization for Nonlinear Model Predictive Control

## A  Expression of Newton System

For future references, we explicitly write out each component of (3). For stage $k$, we let $H_k(z_k, \lambda_k) = \nabla^2_{z_k}(g_k(z_k) - \lambda_k^T f_k(z_k))$, $A_k(z_k) = \nabla^T_{x_k} f_k(z_k)$ and $B_k(z_k) = \nabla^T_{u_k} f_k(z_k)$. Then, we have

$$H^t(\tilde{z}_t, \tilde{\lambda}_t) = \mathrm{diag}\left(H_t, \ldots, H_{M_t-1}, \nabla^2_{x_{M_t}} g_{M_t}(x_{M_t}, \mathbf{0}) + \mu I\right) \tag{12}$$

with $H_k = H_k(z_k, \lambda_k)$ for $k \in [t, M_t - 1]$, and have

$$G^t(\tilde{z}_t) = \begin{pmatrix} I & & & & \\ -A_t & -B_t & I & & \\ & & -A_{t+1} & -B_{t+1} & I \\ & & & & \ddots & \ddots \\ & & & & & -A_{M_t-1} & -B_{M_t-1} & I \end{pmatrix} \tag{13}$$

with $A_k = A_k(z_k)$ and $B_k = B_k(z_k)$. The gradient of Lagrangian $\mathcal{L}^t(\cdot)$ on the right side of (3) can be expressed as

$$\nabla_{\tilde{z}_t} \mathcal{L}^t(\tilde{z}_t, \tilde{\lambda}_t; \bar{x}_t) = \begin{pmatrix} \nabla_{x_t} g_t(z_t) + \lambda_{t-1} - A_t^T(z_t)\lambda_t \\ \nabla_{u_t} g_t(z_t) - B_t^T(z_t)\lambda_t \\ \vdots \\ \nabla_{x_{M_t-1}} g_{M_t-1}(z_{M_t-1}) + \lambda_{M_t-2} - A_{M_t-1}^T(z_{M_t-1})\lambda_{M_t-1} \\ \nabla_{u_{M_t-1}} g_{M_t-1}(z_{M_t-1}) - B_{M_t-1}^T(z_{M_t-1})\lambda_{M_t-1} \\ \nabla_{x_{M_t}} g_{M_t}(x_{M_t}, \mathbf{0}) + \lambda_{M_t-1} + \mu x_{M_t} \end{pmatrix},$$

$$\nabla_{\tilde{\lambda}_t} \mathcal{L}^t(\tilde{z}_t, \tilde{\lambda}_t; \bar{x}_t) = \begin{pmatrix} x_t - \bar{x}_t \\ x_{t+1} - f_t(z_t) \\ \vdots \\ x_{M_t} - f_{M_t-1}(z_t) \end{pmatrix}. \tag{14}$$

We also explicitly write out the gradient of the augmented Lagrangian (5) by

$$\begin{pmatrix} \nabla_{\tilde{z}_t} \mathcal{L}^t_\eta \\ \nabla_{\tilde{\lambda}_t} \mathcal{L}^t_\eta \end{pmatrix} = \begin{pmatrix} I + \eta_2 H^t & \eta_1 (G^t)^T \\ \eta_2 G^t & I \end{pmatrix} \begin{pmatrix} \nabla_{\tilde{z}_t} \mathcal{L}^t \\ \nabla_{\tilde{\lambda}_t} \mathcal{L}^t \end{pmatrix}. \tag{15}$$

## B  Proof of Theorem 4.4

We first have a simple observation: by Assumptions 4.1, 4.2, for any $(\tilde{z}_t, \tilde{\lambda}_t) \in \mathcal{Z} \otimes \Lambda$ (by $(\tilde{z}_t, \tilde{\lambda}_t) \in \mathcal{Z} \otimes \Lambda$ we mean $(\tilde{z}_{k,t}, \tilde{\lambda}_{k,t}) \in \mathcal{Z} \times \Lambda$ for all stages $k$ of the $t$-th subproblem), $\|G^t(\tilde{z}_t)\| \le 1 + 2\Upsilon$, $\|H^t(\tilde{z}_t, \tilde{\lambda}_t)\| \le \Upsilon' + \mu$, and

$$\|\nabla((G^t)^T \nabla_{\tilde{\lambda}_t} \mathcal{L}^t)(\tilde{z}_t, \tilde{\lambda}_t; \bar{x}_t)\| \le \Upsilon', \qquad \|\nabla(H^t \nabla_{\tilde{z}_t} \mathcal{L}^t)(\tilde{z}_t, \tilde{\lambda}_t; \bar{x}_t)\| \le \Upsilon' + \mu^2 \tag{16}$$

for some constant $\Upsilon'$ not depending on $\mu$. This is from the definitions (12)-(14) and noting that only the last block of $H^t$ and the last row of $\nabla_{\tilde{z}_t} \mathcal{L}^t$ contain $\mu$. We can also replace $\Upsilon$ in Assumption 4.2 by $\Upsilon \leftarrow (1 + 2\Upsilon) \vee \Upsilon' \vee \delta$ and require $\mu \ge \Upsilon$. Then we have $\|G^t\| \le \Upsilon$, $\|B^t\| \vee \|H^t\| \le 2\mu$, $\|\nabla((G^t)^T \nabla_{\tilde{\lambda}_t} \mathcal{L}^t)\| \le \Upsilon$, and $\|\nabla(H^t \nabla_{\tilde{z}_t} \mathcal{L}^t)\| \le 2\mu^2$. By the definition of $H^t$ in (12), without loss of generality we let the last block of $B^t$ be $\mu I$.

We then provide a formula for the KKT matrix inverse. We suppress the index $t$ since the results hold for any $t \ge 0$.

**Lemma B.1.** Let $G^T = YK$ where $Y$ has orthonormal columns that span $\mathrm{Im}(G^T)$ and $K$ is a nonsingular square matrix (since $G^T$ has full column rank), and let $Z$ have orthonormal columns that span $\mathrm{Ker}(G)$. If $Z^T B Z$ is invertible, then

$$S := \begin{pmatrix} B & G^T \\ G & \mathbf{0} \end{pmatrix}^{-1} = \begin{pmatrix} S_1 & S_2^T \\ S_2 & S_3 \end{pmatrix}$$

where

$$S_1 = Z(Z^T B Z)^{-1} Z^T,$$
$$S_2 = K^{-1} Y^T (I - B Z (Z^T B Z)^{-1} Z^T),$$
$$S_3 = K^{-1} Y^T (B Z (Z^T B Z)^{-1} Z^T B - B) Y K^{-1}.$$

Under Assumption 4.2, we have $\|S\| \leq 5 \Upsilon^2 \mu^2 / \gamma_{RH}$.

Given Lemma B.1, we apply (3) and (15) and have

$$\begin{pmatrix} \nabla_{\tilde{z}} \mathcal{L}_\eta^0 \\ \nabla_{\tilde{\lambda}} \mathcal{L}_\eta^0 \end{pmatrix}^T \begin{pmatrix} \Delta \tilde{z} \\ \Delta \tilde{\lambda} \end{pmatrix} = - \begin{pmatrix} \nabla_{\tilde{z}} \mathcal{L}^0 \\ \nabla_{\tilde{\lambda}} \mathcal{L}^0 \end{pmatrix}^T \begin{pmatrix} B & G^T \\ G & \mathbf{0} \end{pmatrix}^{-1} \begin{pmatrix} I + \eta_2 H & \eta_1 G^T \\ \eta_2 G & I \end{pmatrix} \begin{pmatrix} \nabla_{\tilde{z}} \mathcal{L}^0 \\ \nabla_{\tilde{\lambda}} \mathcal{L}^0 \end{pmatrix}.$$

By Lemma B.1, we define $W = I - Z(Z^T B Z)^{-1} Z^T B$ and have

$$\begin{pmatrix} B & G^T \\ G & \mathbf{0} \end{pmatrix}^{-1} \begin{pmatrix} I + \eta_2 H & \eta_1 G^T \\ \eta_2 G & I \end{pmatrix}$$
$$= \begin{pmatrix} \eta_2 I + Z(Z^T B Z)^{-1} Z^T \{I + \eta_2 (H - B)\} & W Y (K^{-1})^T \\ K^{-1} Y^T W^T \{I + \eta_2 (H - B)\} & \eta_1 I - K^{-1} Y^T B W Y (K^{-1})^T \end{pmatrix}$$
$$=: W_1 + W_2 + W_3, \tag{17}$$

where

$$W_1 = \begin{pmatrix} \frac{\eta_2}{2} I & \mathbf{0} \\ \mathbf{0} & \frac{\eta_1}{2} I \end{pmatrix},$$
$$W_2 = \begin{pmatrix} \frac{\eta_2}{2} I & W Y (K^{-1})^T \\ K^{-1} Y^T W^T & \frac{\eta_1}{2} I - K^{-1} Y^T B W Y (K^{-1})^T \end{pmatrix},$$
$$W_3 = \begin{pmatrix} Z(Z^T B Z)^{-1} Z^T \{I + \eta_2 (H - B)\} & \mathbf{0} \\ \eta_2 K^{-1} Y^T W^T (H - B) & \mathbf{0} \end{pmatrix}.$$

We deal with each term separately. First, we have

$$\begin{pmatrix} \nabla_{\tilde{z}} \mathcal{L}^0 \\ \nabla_{\tilde{\lambda}} \mathcal{L}^0 \end{pmatrix}^T W_3 \begin{pmatrix} \nabla_{\tilde{z}} \mathcal{L}^0 \\ \nabla_{\tilde{\lambda}} \mathcal{L}^0 \end{pmatrix}$$
$$= \nabla_{\tilde{z}}^T \mathcal{L}^0 Z(Z^T B Z)^{-1} Z^T \nabla_{\tilde{z}} \mathcal{L}^0 + \eta_2 \nabla_{\tilde{z}}^T \mathcal{L}^0 Z(Z^T B Z)^{-1} Z^T (H - B) \nabla_{\tilde{z}} \mathcal{L}^0$$
$$\quad + \eta_2 \nabla_{\tilde{\lambda}}^T \mathcal{L}^0 K^{-1} Y^T W^T (H - B) \nabla_{\tilde{z}} \mathcal{L}^0$$
$$= (\Delta \tilde{z})^T B Z(Z^T B Z)^{-1} Z^T B \Delta \tilde{z} - \eta_2 (\Delta \tilde{z})^T B Z(Z^T B Z)^{-1} Z^T (H - B) \nabla_{\tilde{z}} \mathcal{L}^0$$
$$\quad - \eta_2 (\Delta \tilde{z})^T Y Y^T W^T (H - B) \nabla_{\tilde{z}} \mathcal{L}^0$$
$$= (\Delta \tilde{z})^T B Z(Z^T B Z)^{-1} Z^T B \Delta \tilde{z} - \eta_2 (\Delta \tilde{z})^T (I - W^T)(H - B) \nabla_{\tilde{z}} \mathcal{L}^0$$
$$\quad - \eta_2 (\Delta \tilde{z})^T Y Y^T W^T (H - B) \nabla_{\tilde{z}} \mathcal{L}^0$$
$$= (\Delta \tilde{z})^T B Z(Z^T B Z)^{-1} Z^T B \Delta \tilde{z} - \eta_2 (\Delta \tilde{z})^T (H - B) \nabla_{\tilde{z}} \mathcal{L}^0. \tag{18}$$

Here, the second equality is due to the KKT system (3) and the fact that $GZ = \mathbf{0}$; the third equality is due to the definition of $W$; and the fourth equality is due to $Y Y^T W^T = W^T$. Let us decompose $\Delta \tilde{z} = \Delta \tilde{v} + \Delta \tilde{u}$, where $\Delta \tilde{v} = Z \Delta v$ is a vector in $\text{Im}(Z)$, and $\Delta \tilde{u} = G^T \Delta u$ is a vector in $\text{Im}(G^T)$. Since $G \Delta \tilde{z} = -\nabla_{\tilde{\lambda}} \mathcal{L}^0$ from (3), we know $\Delta u = -(G G^T)^{-1} \nabla_{\tilde{\lambda}} \mathcal{L}^0$ and hence $\Delta \tilde{u} = -G^T (G G^T)^{-1} \nabla_{\tilde{\lambda}} \mathcal{L}^0 = -Y(K^{-1})^T \nabla_{\tilde{\lambda}} \mathcal{L}^0$. Plugging the decomposition into (18), we have

$$\begin{pmatrix} \nabla_{\tilde{z}} \mathcal{L}^0 \\ \nabla_{\tilde{\lambda}} \mathcal{L}^0 \end{pmatrix}^T W_3 \begin{pmatrix} \nabla_{\tilde{z}} \mathcal{L}^0 \\ \nabla_{\tilde{\lambda}} \mathcal{L}^0 \end{pmatrix}$$
$$= (\Delta v)^T Z^T B Z \Delta v - 2(\Delta v)^T Z^T B Y(K^{-1})^T \nabla_{\tilde{\lambda}} \mathcal{L}^0 - \eta_2 (\Delta \tilde{z})^T (H - B) \nabla_{\tilde{z}} \mathcal{L}^0$$
$$\quad + \nabla_{\tilde{\lambda}}^T \mathcal{L}^0 K^{-1} Y^T B Z(Z^T B Z)^{-1} Z^T B Y(K^{-1})^T \nabla_{\tilde{\lambda}} \mathcal{L}^0$$
$$\geq \gamma_{RH} \|\Delta v\|^2 - 4 \mu \Upsilon \|\Delta v\| \|\nabla_{\tilde{\lambda}} \mathcal{L}^0\| - \eta_2 \delta \|\Delta \tilde{z}\| \|\nabla_{\tilde{z}} \mathcal{L}^0\|$$

$$\geq \frac{\gamma_{RH}}{2}\|\Delta\boldsymbol{v}\|^2 - \frac{8\mu^2\Upsilon^2}{\gamma_{RH}}\|\nabla_{\tilde{\boldsymbol{\lambda}}}\mathcal{L}^0\|^2 - \eta_2\delta^2\|\Delta\tilde{\boldsymbol{z}}\|^2 - \frac{\eta_2}{4}\|\nabla_{\tilde{\boldsymbol{z}}}\mathcal{L}^0\|^2$$

$$= \frac{\gamma_{RH}}{2}\|\Delta\boldsymbol{v}\|^2 - \frac{8\mu^2\Upsilon^2}{\gamma_{RH}}\|\nabla_{\tilde{\boldsymbol{\lambda}}}\mathcal{L}^0\|^2 - \eta_2\delta^2(\|\Delta\boldsymbol{v}\|^2 + \|\Delta\tilde{\boldsymbol{u}}\|^2) - \frac{\eta_2}{4}\|\nabla_{\tilde{\boldsymbol{z}}}\mathcal{L}^0\|^2$$

$$\geq \left(\frac{\gamma_{RH}}{2} - \eta_2\delta^2\right)\|\Delta\boldsymbol{v}\|^2 - \left(\frac{8\mu^2\Upsilon^2}{\gamma_{RH}} + \eta_2\delta^2\Upsilon^2\right)\|\nabla_{\tilde{\boldsymbol{\lambda}}}\mathcal{L}^0\|^2 - \frac{\eta_2}{4}\|\nabla_{\tilde{\boldsymbol{z}}}\mathcal{L}^0\|^2,$$

where the second and fifth inequalities are due to Assumption 4.2, which implies $\|K^{-1}\| \leq \Upsilon, \|B\| \vee \|H\| \leq 2\mu$; the third inequality is due to Young's inequality; and the fourth equality is due to $\|\Delta\tilde{\boldsymbol{z}}\|^2 = \|\Delta\tilde{\boldsymbol{v}}\|^2 + \|\Delta\tilde{\boldsymbol{u}}\|^2 = \|\Delta\boldsymbol{v}\|^2 + \|\Delta\tilde{\boldsymbol{u}}\|^2$. Using the above display and supposing

$$\frac{\gamma_{RH}}{2} - \eta_2\delta^2 \geq 0 \iff \eta_2 \leq \frac{\gamma_{RH}}{2\delta^2}, \tag{19}$$

we further have

$$\begin{pmatrix}\nabla_{\tilde{\boldsymbol{z}}}\mathcal{L}^0 \\ \nabla_{\tilde{\boldsymbol{\lambda}}}\mathcal{L}^0\end{pmatrix}^T W_3 \begin{pmatrix}\nabla_{\tilde{\boldsymbol{z}}}\mathcal{L}^0 \\ \nabla_{\tilde{\boldsymbol{\lambda}}}\mathcal{L}^0\end{pmatrix} \geq -\left(\frac{8\mu^2\Upsilon^2}{\gamma_{RH}} + \frac{\gamma_{RH}\Upsilon^2}{2}\right)\|\nabla_{\tilde{\boldsymbol{\lambda}}}\mathcal{L}^0\|^2 - \frac{\eta_2}{4}\|\nabla_{\tilde{\boldsymbol{z}}}\mathcal{L}^0\|^2$$

$$\geq -\frac{9\mu^2\Upsilon^2}{\gamma_{RH}}\|\nabla_{\tilde{\boldsymbol{\lambda}}}\mathcal{L}^0\|^2 - \frac{\eta_2}{4}\|\nabla_{\tilde{\boldsymbol{z}}}\mathcal{L}^0\|^2. \tag{20}$$

Let us now deal with $W_2$. By Schur complement, in order to show $W_2 \succeq \boldsymbol{0}$, we only have to let

$$\frac{\eta_1}{2}I - K^{-1}Y^T BWY(K^{-1})^T - \frac{2}{\eta_2}K^{-1}Y^T W^T WY(K^{-1})^T \succeq \boldsymbol{0}. \tag{21}$$

Note that $-K^{-1}Y^T BWY(K^{-1})^T \succeq -K^{-1}Y^T BY(K^{-1})^T$ and

$$\|K^{-1}Y^T BY(K^{-1})^T + \frac{2}{\eta_2}K^{-1}Y^T W^T WY(K^{-1})^T\| \leq 2\mu\Upsilon^2 + \frac{2\Upsilon^2}{\eta_2}\|W\|^2$$

$$\leq 2\mu\Upsilon^2 + \frac{2\Upsilon^2}{\eta_2}\left(1 + \frac{2\mu}{\gamma_{RH}}\right)^2 = 2\mu\Upsilon^2 + \frac{2\Upsilon^2}{\eta_2} + \frac{8\mu\Upsilon^2}{\eta_2\gamma_{RH}} + \frac{8\mu^2\Upsilon^2}{\eta_2\gamma_{RH}^2}$$

$$\leq \frac{12\mu\Upsilon^2}{\eta_2\gamma_{RH}} + \frac{8\mu^2\Upsilon^2}{\eta_2\gamma_{RH}^2} \leq \frac{16\mu^2\Upsilon^2}{\eta_2\gamma_{RH}^2},$$

where the fifth inequality supposes $\gamma_{RH} \leq \sqrt{2}\delta$ (without loss of generality, since $\delta$ is upper bound and $\gamma_{RH}$ is lower bound in Assumption 4.2) so that $\eta_2\gamma_{RH} \leq 1$; and the last inequality uses $\mu \geq 2\gamma_{RH}$. Thus, we only have to let

$$\frac{\eta_1}{2} \geq \frac{16\mu^2\Upsilon^2}{\eta_2\gamma_{RH}^2} \iff \eta_1\eta_2 \geq \frac{32\mu^2\Upsilon^2}{\gamma_{RH}^2}, \tag{22}$$

then (21) is satisfied and $W_2 \succeq \boldsymbol{0}$. Combining (17), (20), and noting that $W_1$ is a diagonal matrix, we obtain that under (19) and (22),

$$\begin{pmatrix}\nabla_{\tilde{\boldsymbol{z}}}\mathcal{L}_\eta^0 \\ \nabla_{\tilde{\boldsymbol{\lambda}}}\mathcal{L}_\eta^0\end{pmatrix}^T \begin{pmatrix}\Delta\tilde{\boldsymbol{z}} \\ \Delta\tilde{\boldsymbol{\lambda}}\end{pmatrix} \leq -\begin{pmatrix}\nabla_{\tilde{\boldsymbol{z}}}\mathcal{L}^0 \\ \nabla_{\tilde{\boldsymbol{\lambda}}}\mathcal{L}^0\end{pmatrix}^T \begin{pmatrix}\frac{\eta_2}{4} & \boldsymbol{0} \\ \boldsymbol{0} & \frac{\eta_1}{2} - \frac{9\mu^2\Upsilon^2}{\gamma_{RH}}\end{pmatrix} \begin{pmatrix}\nabla_{\tilde{\boldsymbol{z}}}\mathcal{L}^0 \\ \nabla_{\tilde{\boldsymbol{\lambda}}}\mathcal{L}^0\end{pmatrix}.$$

Using $\gamma_{RH} \leq 6\mu\delta\Upsilon$, we can easily check that, as long as $\eta = (\eta_1, \eta_2)$ satisfies

$$\eta_1 \geq \frac{25\mu^2\Upsilon^2}{\gamma_{RH}} =: \tau_1, \quad \eta_2 \leq \frac{\gamma_{RH}}{2\delta^2} =: \tau_2, \quad \eta_1\eta_2 \geq \frac{32\mu^2\Upsilon^2}{\gamma_{RH}^2} =: \tau_3,$$

we have

$$\begin{pmatrix}\nabla_{\tilde{\boldsymbol{z}}}\mathcal{L}_\eta^0 \\ \nabla_{\tilde{\boldsymbol{\lambda}}}\mathcal{L}_\eta^0\end{pmatrix}^T \begin{pmatrix}\Delta\tilde{\boldsymbol{z}} \\ \Delta\tilde{\boldsymbol{\lambda}}\end{pmatrix} \leq -\frac{\eta_2}{4}\left\|\begin{pmatrix}\nabla_{\tilde{\boldsymbol{z}}}\mathcal{L}^0 \\ \nabla_{\tilde{\boldsymbol{\lambda}}}\mathcal{L}^0\end{pmatrix}\right\|^2.$$

This completes the proof of the first part of the statement. For the second part of the statement. We note that $\eta_2^0 = 1$ and each While loop decreases $\eta_2^0$ by $\rho$. Thus, to satisfy $\eta_2 \leq \tau_2$, the number of the

required While loop iterations $\mathcal{T}$ only need satisfy $\rho^{\mathcal{T}} \geq 1/\tau_2$. For the similar reason, we require $\rho^{\mathcal{T}} \geq \tau_3/\mu^2$ and $\rho^{\mathcal{T}} \geq \sqrt{\tau_1/\mu^2}$. Combining them together, we know if $\mathcal{T}$ satisfies

$$\rho^{\mathcal{T}} \geq \left( \frac{1}{\tau_2} \vee \frac{\tau_3}{\mu^2} \vee \sqrt{\frac{\tau_1}{\mu^2}} \right) = \frac{32\Upsilon^2}{\gamma_{RH}^2},$$

then no other iterations will go into the While loop again. Thus, we know $\rho^{\mathcal{T}} \leq \frac{32\Upsilon^2 \rho}{\gamma_{RH}^2}$. Moreover,

$$\bar{\eta}_2 = 1/\rho^{\mathcal{T}} \geq \frac{\gamma_{RH}}{32\Upsilon^2\rho}, \quad \text{and} \quad \bar{\eta}_1 = \mu^2(\rho^{\mathcal{T}})^2 \leq \frac{32^2\rho^2\mu^2\Upsilon^4}{\gamma_{RH}^4}.$$

This completes the second part of the statement.

## C   Proof of Lemma B.1

We note that $YY^T + ZZ^T = I$. Thus, $YY^T(I - BZ(Z^TBZ)^{-1}Z^T) = I - BZ(Z^TBZ)^{-1}Z^T$. Using this observation, the formula of $S$ can be verified directly by checking $SS^{-1} = I$. Moreover, under Assumption 4.2, we know

$$\|(Z^TBZ)^{-1}\| \leq 1/\gamma_{RH}, \quad \|K^{-1}\| \leq \Upsilon, \quad \text{and} \quad \|B\| \leq 2\mu.$$

Therefore,

$$\|S\| \leq \|S_1\| + 2\|S_2\| + \|S_3\| \leq \frac{1}{\gamma_{RH}} + 2\Upsilon(1 + \frac{2\mu}{\gamma_{RH}}) + \Upsilon^2 \left( \frac{4\mu^2}{\gamma_{RH}} + 2\mu \right).$$

Without loss of generality, we suppose $\Upsilon \geq 4$ and $\mu \geq 2(\gamma_{RH} + 1)$. Then

$$\|S\| \leq \frac{1}{\gamma_{RH}} + \frac{6\Upsilon\mu}{\gamma_{RH}} + 2\mu\Upsilon^2 + \frac{4\Upsilon^2\mu^2}{\gamma_{RH}} \leq \frac{\Upsilon^2\mu^2}{\gamma_{RH}} + \frac{4\Upsilon^2\mu^2}{\gamma_{RH}} \leq \frac{5\Upsilon^2\mu^2}{\gamma_{RH}}.$$

This completes the proof.

## D   Proof of Theorem 4.5

We drop off the index $t$ for simplicity. By the definition of $\mathcal{L}_\eta(\cdot)$ in (5), we have

$$\nabla^2 \mathcal{L}_\eta(\tilde{z}, \tilde{\lambda}; \bar{x}) = \begin{pmatrix} H + \eta_2 \nabla_{\tilde{z}}(H\nabla_{\tilde{z}}\mathcal{L}) + \eta_1 \nabla_{\tilde{z}}(G^T\nabla_{\tilde{\lambda}}\mathcal{L}) & \eta_2 \nabla_{\tilde{\lambda}}^T(H\nabla_{\tilde{z}}\mathcal{L}) + G^T \\ \eta_2 \nabla_{\tilde{\lambda}}(H\nabla_{\tilde{z}}\mathcal{L}) + G & \eta_2 GG^T \end{pmatrix}.$$

Using Assumption 4.2, (16), and Theorem 4.4, we know

$$\|\nabla^2 \mathcal{L}_\eta(\tilde{z}, \tilde{\lambda}; \bar{x})\| \leq 4\bar{\eta}_1\Upsilon \leq \frac{32^2\rho^2\mu^2\Upsilon^5}{\gamma_{RH}} =: \mu^2\Upsilon'.$$

Therefore, by Taylor expansion

$$\mathcal{L}_{\bar{\eta}}^1 \leq \mathcal{L}_{\bar{\eta}}^0 + \alpha \begin{pmatrix} \nabla_{\tilde{z}}\mathcal{L}_{\bar{\eta}}^0 \\ \nabla_{\tilde{\lambda}}\mathcal{L}_{\bar{\eta}}^0 \end{pmatrix}^T \begin{pmatrix} \Delta\tilde{z} \\ \Delta\tilde{\lambda} \end{pmatrix} + \frac{\mu^2\Upsilon'\alpha^2}{2} \left\| \begin{pmatrix} \Delta\tilde{z} \\ \Delta\tilde{\lambda} \end{pmatrix} \right\|^2. \tag{23}$$

Moreover, by Lemma B.1 and the condition (7), we further have

$$\left\| \begin{pmatrix} \Delta\tilde{z} \\ \Delta\tilde{\lambda} \end{pmatrix} \right\|^2 \leq \frac{25\mu^4\Upsilon^4}{\gamma_{RH}^2} \left\| \begin{pmatrix} \nabla_{\tilde{z}}\mathcal{L}^0 \\ \nabla_{\tilde{\lambda}}\mathcal{L}^0 \end{pmatrix} \right\|^2 \leq -\frac{100\mu^4\Upsilon^4}{\bar{\eta}_2\gamma_{RH}} \begin{pmatrix} \nabla_{\tilde{z}}\mathcal{L}_{\bar{\eta}}^0 \\ \nabla_{\tilde{\lambda}}\mathcal{L}_{\bar{\eta}}^0 \end{pmatrix}^T \begin{pmatrix} \Delta\tilde{z} \\ \Delta\tilde{\lambda} \end{pmatrix}.$$

Plugging the above display into (23),

$$\mathcal{L}_{\bar{\eta}}^1 \leq \mathcal{L}_{\bar{\eta}}^0 + \alpha \left( 1 - \frac{50\mu^6\Upsilon'\Upsilon^4}{\bar{\eta}_2\gamma_{RH}}\alpha \right) \begin{pmatrix} \nabla_{\tilde{z}}\mathcal{L}_{\bar{\eta}}^0 \\ \nabla_{\tilde{\lambda}}\mathcal{L}_{\bar{\eta}}^0 \end{pmatrix}^T \begin{pmatrix} \Delta\tilde{z} \\ \Delta\tilde{\lambda} \end{pmatrix}.$$

Thus, as long as

$$1 - \frac{50\mu^6\Upsilon'\Upsilon^4}{\bar{\eta}_2\gamma_{RH}}\alpha \geq \beta \iff \alpha \leq \frac{(1-\beta)\bar{\eta}_2\gamma_{RH}}{50\mu^6\Upsilon'\Upsilon^4} \Longleftarrow \alpha \leq \frac{(1-\beta)\gamma_{RH}^2}{32 \cdot 50\mu^6\Upsilon'\Upsilon^6} =: \bar{\alpha}',$$

then Armijo condition (6) is satisfied. Thus, if we use backtracking line search, the selected stepsize $\alpha \geq \nu\bar{\alpha}' =: \bar{\alpha}$ for some $\nu \in (0, 1)$. Moreover, by Armijo condition,

$$\mathcal{L}_{\bar{\eta}}^1 \leq \mathcal{L}_{\bar{\eta}}^0 + \alpha\beta \begin{pmatrix} \nabla_{\tilde{z}}\mathcal{L}_{\bar{\eta}}^0 \\ \nabla_{\tilde{\lambda}}\mathcal{L}_{\bar{\eta}}^0 \end{pmatrix}^T \begin{pmatrix} \Delta\tilde{z} \\ \Delta\tilde{\lambda} \end{pmatrix} \leq \mathcal{L}_{\bar{\eta}}^0 - \frac{\bar{\eta}_2\bar{\alpha}\beta}{4} \left\| \begin{pmatrix} \nabla_{\tilde{z}}\mathcal{L}^0 \\ \nabla_{\tilde{\lambda}}\mathcal{L}^0 \end{pmatrix} \right\|^2.$$

This completes the proof.

# E  Proof of Lemma 4.6

By the definition (5), we know

$$\mathcal{L}_{\tilde{\eta}}^{t,1} - \mathcal{L}_{\tilde{\eta}}^{t+1,0}$$

$$= \mathcal{L}^{t,1} - \mathcal{L}^{t+1,0} + \frac{\bar{\eta}_1}{2}\left(\left\|\nabla_{\tilde{\boldsymbol{\lambda}}_t}\mathcal{L}^{t,1}\right\|^2 - \left\|\nabla_{\tilde{\boldsymbol{\lambda}}_{t+1}}\mathcal{L}^{t+1,0}\right\|^2\right) + \frac{\bar{\eta}_2}{2}\left(\left\|\nabla_{\tilde{\boldsymbol{z}}_t}\mathcal{L}^{t,1}\right\|^2 - \left\|\nabla_{\tilde{\boldsymbol{z}}_{t+1}}\mathcal{L}^{t+1,0}\right\|^2\right)$$

$$=: Term_1 + Term_2 + Term_3. \tag{24}$$

Let us deal with each term separately. For $Term_1$, we apply the definition of Lagrangian function, the transition (8), and the fact that $g(\boldsymbol{0}, \boldsymbol{0}) = 0$. Then

$$\mathcal{L}^{t+1,0} = \sum_{k=t+1}^{M_t}\left\{g_k(\boldsymbol{z}_{k,t+1}^0) + (\boldsymbol{\lambda}_{k-1,t+1}^0)^T\boldsymbol{x}_{k,t+1}^0 - (\boldsymbol{\lambda}_{k,t+1}^0)^Tf_k(\boldsymbol{z}_{k,t+1}^0)\right\} + g_{M_t+1}(\boldsymbol{x}_{M_t+1,t+1}^0, \boldsymbol{0})$$

$$+ \frac{\mu}{2}\|\boldsymbol{x}_{M_t+1,t+1}^0\| + (\boldsymbol{\lambda}_{M_t,t+1}^0)^T\boldsymbol{x}_{M_t+1,t+1}^0 - (\boldsymbol{\lambda}_{t,t+1}^0)^T\bar{\boldsymbol{x}}_{t+1}$$

$$= \sum_{k=t+1}^{M_t-1}\left\{g_k(\boldsymbol{z}_{k,t}^1) + (\boldsymbol{\lambda}_{k-1,t}^1)^T\boldsymbol{x}_{k,t}^1 - (\boldsymbol{\lambda}_{k,t}^1)^Tf_k(\boldsymbol{z}_{k,t}^1)\right\} + g_{M_t}(\boldsymbol{x}_{M_t,t}^1, \boldsymbol{0}) + (\boldsymbol{\lambda}_{M_t-1,t}^1)^T\boldsymbol{x}_{M_t,t}^1$$

$$- (\boldsymbol{\lambda}_{t,t}^1)^Tf_t(\boldsymbol{z}_{t,t}^1).$$

Using the above display, we further have

$$Term_1 = \mathcal{L}^{t,1} - \mathcal{L}^{t+1,0}$$

$$= \sum_{k=t}^{M_t-1}\left\{g_k(\boldsymbol{z}_{k,t}^1) + (\boldsymbol{\lambda}_{k-1,t}^1)^T\boldsymbol{x}_{k,t}^1 - (\boldsymbol{\lambda}_{k,t}^1)^Tf_k(\boldsymbol{z}_{k,t}^1)\right\} + g_{M_t}(\boldsymbol{x}_{M_t,t}^1, \boldsymbol{0}) + \frac{\mu}{2}\|\boldsymbol{x}_{M_t,t}^1\|^2$$

$$+ (\boldsymbol{\lambda}_{M_t-1,t}^1)^T\boldsymbol{x}_{M_t,t}^1 - (\boldsymbol{\lambda}_{t-1,t}^1)^T\bar{\boldsymbol{x}}_t - \mathcal{L}^{t+1,0}$$

$$= g_t(\boldsymbol{z}_{t,t}^1) + (\boldsymbol{\lambda}_{t-1,t}^1)^T(\boldsymbol{x}_{t,t}^1 - \bar{\boldsymbol{x}}_t) + \frac{\mu}{2}\|\boldsymbol{x}_{M_t,t}^1\|^2$$

$$\geq -\|\boldsymbol{\lambda}_{t-1,t}^1\|\|\boldsymbol{x}_{t,t}^1 - \bar{\boldsymbol{x}}_t\| + \frac{\mu}{2}\|\boldsymbol{x}_{M_t,t}^1\|^2$$

$$\geq -C\|\boldsymbol{x}_{t,t}^1 - \bar{\boldsymbol{x}}_t\|^2 + \frac{\mu}{2}\|\boldsymbol{x}_{M_t,t}^1\|^2, \tag{25}$$

where the last inequality is due to Assumption 4.3(ii). For $Term_2$, we apply the formula (14) and the transition (8). We have

$$\left\|\nabla_{\tilde{\boldsymbol{\lambda}}_{t+1}}\mathcal{L}^{t+1,0}\right\|^2 = \sum_{k=t+1}^{M_t}\|\boldsymbol{x}_{k+1,t+1}^0 - f_k(\boldsymbol{z}_{k,t+1}^0)\|^2 + \|\boldsymbol{x}_{t+1,t+1}^0 - \bar{\boldsymbol{x}}_{t+1}\|^2$$

$$= \sum_{k=t+1}^{M_t-1}\|\boldsymbol{x}_{k+1,t}^1 - f_k(\boldsymbol{z}_{k,t}^1)\|^2 + \|f_{M_t}(\boldsymbol{x}_{M_t,t}^1, \boldsymbol{0})\|^2 + \|\boldsymbol{x}_{t+1,t}^1 - f_t(\boldsymbol{z}_{t,t}^1)\|^2$$

$$= \sum_{k=t}^{M_t-1}\|\boldsymbol{x}_{k+1,t}^1 - f_k(\boldsymbol{z}_{k,t}^1)\|^2 + \|f_{M_t}(\boldsymbol{x}_{M_t,t}^1, \boldsymbol{0})\|^2.$$

Using the above display, we further have

$$Term_2 = \frac{\bar{\eta}_1}{2}\left(\left\|\nabla_{\tilde{\boldsymbol{\lambda}}_t}\mathcal{L}^{t,1}\right\|^2 - \left\|\nabla_{\tilde{\boldsymbol{\lambda}}_{t+1}}\mathcal{L}^{t+1,0}\right\|^2\right) = \frac{\bar{\eta}_1}{2}\|\boldsymbol{x}_{t,t}^1 - \bar{\boldsymbol{x}}_t\|^2 - \frac{\bar{\eta}_1}{2}\|f_{M_t}(\boldsymbol{x}_{M_t,t}^1, \boldsymbol{0})\|^2$$

$$\geq \frac{\bar{\eta}_1}{2}\|\boldsymbol{x}_{t,t}^1 - \bar{\boldsymbol{x}}_t\|^2 - \frac{\bar{\eta}_1\Upsilon^2}{2}\|\boldsymbol{x}_{M_t,t}^1\|^2, \tag{26}$$

where the last inequality is due to Assumption 4.2. Last, for $Term_3$, we apply the formula (14) and the transition (8). We have

$$\|\nabla_{\tilde{z}_{t+1}}\mathcal{L}^{t+1,0}\|^2 = \sum_{k=t+1}^{M_t}\left\|\begin{pmatrix}\nabla_{\boldsymbol{x}_k}g_k(\boldsymbol{z}_{k,t+1}^0)+\boldsymbol{\lambda}_{k-1,t+1}^0-A_k^T(\boldsymbol{z}_{k,t+1}^0)\boldsymbol{\lambda}_{k,t+1}^0\\\nabla_{\boldsymbol{u}_k}g_k(\boldsymbol{z}_{k,t+1}^0)-B_k^T(\boldsymbol{z}_{k,t+1}^0)\boldsymbol{\lambda}_{k,t+1}^0\end{pmatrix}\right\|^2$$
$$+\|\nabla_{\boldsymbol{x}_{M_t+1}}g_{M_t+1}(\boldsymbol{x}_{M_t+1,t+1}^0,\mathbf{0})+\boldsymbol{\lambda}_{M_t,t+1}^0+\mu\boldsymbol{x}_{M_t+1,t+1}^0\|^2$$
$$=\sum_{k=t+1}^{M_t-1}\left\|\begin{pmatrix}\nabla_{\boldsymbol{x}_k}g_k(\boldsymbol{z}_{k,t}^1)+\boldsymbol{\lambda}_{k-1,t}^1-A_k^T(\boldsymbol{z}_{k,t}^1)\boldsymbol{\lambda}_{k,t}^1\\\nabla_{\boldsymbol{u}_k}g_k(\boldsymbol{z}_{k,t}^1)-B_k^T(\boldsymbol{z}_{k,t}^1)\boldsymbol{\lambda}_{k,t}^1\end{pmatrix}\right\|^2$$
$$+\left\|\begin{pmatrix}\nabla_{\boldsymbol{x}_{M_t}}g_{M_t}(\boldsymbol{x}_{M_t,t}^1,\mathbf{0})+\boldsymbol{\lambda}_{M_t-1,t}^1\\\nabla_{\boldsymbol{u}_{M_t}}g_{M_t}(\boldsymbol{x}_{M_t,t}^1,\mathbf{0})\end{pmatrix}\right\|^2.$$

Using the above display, we further have

$$Term_3 = \frac{\bar{\eta}_2}{2}\left(\|\nabla_{\tilde{z}_t}\mathcal{L}^{t,1}\|^2-\|\nabla_{\tilde{z}_{t+1}}\mathcal{L}^{t+1,0}\|^2\right)$$
$$\geq\frac{\bar{\eta}_2}{2}\left(\|\nabla_{\boldsymbol{x}_{M_t}}g_{M_t}(\boldsymbol{x}_{M_t,t}^1,\mathbf{0})+\boldsymbol{\lambda}_{M_t-1,t}^1+\mu\boldsymbol{x}_{M_t,t}^1\|^2-\left\|\begin{pmatrix}\nabla_{\boldsymbol{x}_{M_t}}g_{M_t}(\boldsymbol{x}_{M_t,t}^1,\mathbf{0})+\boldsymbol{\lambda}_{M_t-1,t}^1\\\nabla_{\boldsymbol{u}_{M_t}}g_{M_t}(\boldsymbol{x}_{M_t,t}^1,\mathbf{0})\end{pmatrix}\right\|^2\right)$$
$$\geq\frac{\bar{\eta}_2(\mu^2-\Upsilon^2)}{2}\|\boldsymbol{x}_{M_t,t}^1\|^2+\bar{\eta}_2\mu\left((\boldsymbol{x}_{M_t,t}^1)^T\nabla_{\boldsymbol{x}_{M_t}}g_{M_t}(\boldsymbol{x}_{M_t,t}^1,\mathbf{0})+(\boldsymbol{x}_{M_t,t}^1)^T\boldsymbol{\lambda}_{M_t-1,t}^1\right)$$
$$\geq\frac{\bar{\eta}_2(\mu^2-\Upsilon^2-2\mu\Upsilon)}{2}\|\boldsymbol{x}_{M_t,t}^1\|^2+\bar{\eta}_2\mu(\boldsymbol{x}_{M_t,t}^1)^T\boldsymbol{\lambda}_{M_t-1,t}^1,$$

where the second inequality is due to the definition of $\nabla_{\tilde{z}_t}\mathcal{L}^{t,1}$; and the third and the fourth inequalities are due to Assumption 4.2, which implies $\|\nabla_{\boldsymbol{z}_{M_t}}g_{M_t}(\boldsymbol{x}_{M_t,t}^1,\mathbf{0})\|\leq\Upsilon\|\boldsymbol{x}_{M_t,t}^1\|$. Noting that $\boldsymbol{\lambda}_{M_t-1,t}^0=\mathbf{0}$ and, by (3),

$$\mu\Delta\tilde{\boldsymbol{x}}_{M_t,t}+\Delta\tilde{\boldsymbol{\lambda}}_{M_t-1,t}=-\left(\nabla_{\boldsymbol{x}_{M_t}}g_{M_t}(\boldsymbol{x}_{M_t,t}^0,\mathbf{0})+\boldsymbol{\lambda}_{M_t-1,t}^0+\mu\boldsymbol{x}_{M_t,t}^0\right)=\mathbf{0},$$

we then have

$$(\boldsymbol{x}_{M_t,t}^1)^T\boldsymbol{\lambda}_{M_t-1,t}^1=-\alpha_t\mu(\boldsymbol{x}_{M_t,t}^1)^T\Delta\tilde{\boldsymbol{x}}_{M_t,t}=-\mu\|\boldsymbol{x}_{M_t,t}^1\|^2.$$

Suppose $\mu\geq4\Upsilon$, then $\mu^2-\Upsilon^2-2\mu\Upsilon\geq\mu^2/2$. Together with the above three displays,

$$Term_3\geq-\bar{\eta}_2\mu^2\|\boldsymbol{x}_{M_t,t}^1\|^2. \tag{27}$$

Combining (24), (25), (26), and (27), and noting that $\bar{\eta}_2\mu^2\leq\mu^2\leq\bar{\eta}_1\Upsilon^2/2$, we have

$$\mathcal{L}_{\bar{\eta}}^{t,1}-\mathcal{L}_{\bar{\eta}}^{t+1,0}\geq\left(\frac{\bar{\eta}_1}{2}-C\right)\|\boldsymbol{x}_{t,t}^1-\bar{\boldsymbol{x}}_t\|^2+\left(\frac{\mu}{2}-\frac{\bar{\eta}_1\Upsilon^2}{2}-\bar{\eta}_2\mu^2\right)\|\boldsymbol{x}_{M_t,t}^1\|^2$$
$$\geq\left(\frac{\mu^2}{2}-C\right)\|\boldsymbol{x}_{t,t}^1-\bar{\boldsymbol{x}}_t\|^2-\bar{\eta}_1\Upsilon^2\|\boldsymbol{x}_{M_t,t}^1\|^2\geq-\bar{\eta}_1\Upsilon^2\|\boldsymbol{x}_{M_t,t}^1\|^2,$$

where the last inequality holds if $C\leq\mu^2/2$. By Lemma B.1, Theorem 4.4 and Assumption 4.3(i),

$$\bar{\eta}_1\Upsilon^2\|\boldsymbol{x}_{M_t,t}^1\|^2\leq\frac{32^2\rho^2\Upsilon^6}{\gamma_{RH}^4}\mu^2\alpha_t^2\|\Delta\tilde{\boldsymbol{x}}_{M_t,t}\|^2=\frac{32^2\rho^2\Upsilon^6}{\gamma_{RH}^4}\alpha_t^2\|\Delta\tilde{\boldsymbol{\lambda}}_{M_t-1,t}\|^2$$
$$\leq\frac{32^2\rho^2\Upsilon^6}{\gamma_{RH}^4}c^2\|(\Delta\tilde{\boldsymbol{z}}_t,\Delta\tilde{\boldsymbol{\lambda}}_t)\|^2\leq\frac{32^2\rho^2\Upsilon^6c^2}{\gamma_{RH}^4}\|\nabla\mathcal{L}^{t,0}\|^2.$$

We require

$$\frac{32^2\rho^2\Upsilon^6c^2}{\gamma_{RH}^4}\leq\frac{\bar{\eta}_2\bar{\alpha}\beta}{8}\Longleftarrow\frac{32^2\rho^2\Upsilon^6c^2}{\gamma_{RH}^4}\leq\frac{\beta\gamma_{RH}\bar{\alpha}}{8\times32\rho\Upsilon^2}$$
$$\Longleftarrow\frac{32^2\rho^2\Upsilon^6c^2}{\gamma_{RH}^4}\leq\frac{\beta(1-\beta)\gamma_{RH}^3}{20^2\times32^2\rho\mu^6\Upsilon'\Upsilon^8}$$
$$\Longleftrightarrow c^2\lesssim\frac{\gamma_{RH}^2}{\kappa^6}.$$

where the first implication is due to Theorem 4.4; and the second implication is due to Theorem 4.5. Then, we have

$$\mathcal{L}_{\bar{\eta}}^{t,1} - \mathcal{L}_{\bar{\eta}}^{t+1,0} \geq -\frac{\bar{\eta}_2 \bar{\alpha} \beta}{8} \|\nabla \mathcal{L}^{t,0}\|^2.$$

This completes the proof.

## F Proof of Theorem 4.7

Summing over $t$ from $\tau$ to $\infty$ on both sides of (11), we have

$$\frac{\bar{\eta}_2 \bar{\alpha} \beta}{8} \sum_{t=\tau}^{\infty} \|\nabla \mathcal{L}^{t,0}\|^2 \leq \mathcal{L}_{\bar{\eta}}^{0,\tau} - \min_{\mathcal{Z} \otimes \Lambda} \mathcal{L}_{\bar{\eta}}(\tilde{z}, \tilde{\lambda}; \bar{x}) < \infty.$$

Thus, $\|\nabla \mathcal{L}^{t,0}\|^2 \to 0$ as $t \to \infty$. We complete the proof.