# OpenReview forum: "Global Convergence of Online Optimization for Nonlinear Model Predictive Control"
_NeurIPS.cc/2021/Conference — NeurIPS 2021 Poster_

### Official Review · Reviewer_fJhh · 2021-07-14

**Rating:** 6
**Confidence:** 4

**Summary:**

This paper gives an algorithm for nonlinear MPC based on augmented Lagrangians, and demonstrates that the iterates converge globally to a critical point of the KKT equations. The key insight is a rule for tuning the regularization parameters of the augmented Lagrangian that ensures that each algorithm step reduces the KKT residuals sufficiently to overcome any increase that would arise due to taking a step in the dynamics. As a result, after alternating between algorithm steps and dynamics steps for several iterations, all iterates produced by the algorithm will be near critical points.

**Ethical Concerns:**

No concerns.

**Limitations And Societal Impact:**

The authors have addressed some of the basic limitations of the work in the conclusion. There does not appear to be any obvious negative societal impact.

**Main Review:**

The paper is well written, the topic is interesting, and the theory appears to be sound. However, I do have several concerns.

Fit
My most basic concern with this work is its fit with NeurIPs. This appears to be a work on model predictive control, with very little overlap with the machine learning community. While many control-related topics such as reinforcement learning and system identification have broad overlap with NeurIPs, this work does not. I think a controls conference or journal would be a much better fit.

Significance
It is not clear what global convergence really implies in this context. The KKT residuals go to zero, but what does this mean from a control perspective? It appears that the after many iterates, the solution will at least be nearly feasible. But it is unclear whether the solution is converging to a local minimum.

Along with this, the assumption that the state and actions stay within a compact set place some stringent requirement a priori that the MPC method stabilizes the system.

Edit after Response: The general consensus seems to be that this is within the scope of NuerIPs, which was my main concern. I do think the work has merit, but there are some limitations, such as closed-loop stability, which are not addressed.

**Time Spent Reviewing:**

2

---

> ### Author Response · Authors · 2021-08-05
> **The important message of the paper may be missed by the reviewer.**
>
> We thank the reviewer for the comments, although we respectfully disagree with them all.
>
> In summary, this paper suits NeruIPS well and does not need control background, but only optimization knowledge. Further, we consider nonlinear (certainly nonconvex) problems and only impose global assumptions to study the global behavior of an online scheme. Converging to local minimum requires additional local assumptions [37]. Judging the significance only based on stationarity completely misses the contribution of the paper: we show that the global convergence is achievable for online iterates with varying subproblems, where only a single Newton step is performed for each subproblem. Finally, the compactness condition is mild, not restrictive. It is so common in the existing literature (see [37, 2]).
>
> We explain in details as follows.
>
> ***1***, This paper studies a problem that intersects online optimization, nonlinear (certainly nonconvex) optimization, control and planning, which are all valued topics in NeurIPS. Our audience include optimizer and (at least model-based) RL people, who gathered in NeurIPS community. It is not fair to exclude our paper from these potential audience. For example, a similar MPC problem with local analysis is studied in [30] in NeurIPS; so online MPC is definitely valuable within NeurIPS scoop.
>
> Furthermore, this paper does not require control background because our techniques are all based on optimization. In an abstract form, our paper studies a parametric optimization, where the problem is parameterized by a varying index, and one Newton step is performed at each index. We answer the question that whether global convergence is possible in such a setup. This question may interest optimization people. In addition, we can extend our augmented Lagrangian technique to model-based RL, which may interest RL people as well.
>
> ***2***, We are sorry but we would argue that our result is significant even we converge to a KKT point. In some sense, it is not fair to require our procedure to always converge to a local minimum, just like requiring SGD to converge to a local minimum for nonconvex objective. Please see the response to the first reviewer. Converging to a local minimum is desirable but not the goal of the paper. In general, converging to a local minimum requires two steps: first the iterate converges to a stationary point; second, under additional assumptions, the stationary point that the iterate converges to is a local minimum. This paper only studies the first step; while the second step has been originally studied in [10].
>
> Converging to a KKT point is the first, necessary result for studying the convergence of an algorithm. Converging to a local minimum is only possible under additional local assumptions. Under our global assumptions, converging to a KKT point is the best we can achieve. Please see [37, Chapters 3, 18] (e.g. Theorem 18.3) for the meaning of global convergence of nonlinear problems. All global convergence in nonlinear literature [37] is converging to a KKT point. We strongly believe that judging the significance based on "KKT v.s. local minimum" is not fair and not suitable regarding the goal of the paper. See reviewers 1 and 2 for summary of significance.
>
>
> ***3***, Again, we are sorry but have to oppose this point. The compact assumption is not restrictive at all. It just excludes the case where KKT point is at infinity point. Such an assumption is very common in the literature. See the same setup in [37, Theorem 18.3] and whole Chapter 4 (e.g. Proposition 4.15) of [2]. We emphasize that the compact set need not be small, so the assumption completely has no relation to stability of MPC.
>
>
> ***In summary***, we sincerely hope the reviewer could re-evaluate the paper based on the problem challenge, novelty, and significance. We wholeheartedly believe that convergence of online iterates, MPC, nonlinear optimization are all in NeurIPS scoop. Our global convergence result is novel, interesting, and important to the community. The compactness condition is standard in the literature and does not exclude important applications.

---

> > ### Comment · Reviewer_fJhh · 2021-08-25
> > **I see the merit. Stability of closed-loop is still not addressed.**
> >
> > Based on the other reviewers points and the responses, I will raise my score a bit. Others all agree that this is within the scope of NeurIPS. I do agree, as well that converging to a KKT point is valuable. I was just wondering if the results could be made stronger.
> >
> > As to the point of stability, I mean stability of the closed-loop system. Consider the following MPC problem, which is a special case of the problem considered here:
> >
> > \begin{align*}
> > &\min && x_t^2 +u_t^2 + x_{t+1}^2 \\
> > & \textrm{subject to} && x_{t+1} = ax_t + u_t \\
> > &&& x_t = \bar x_t
> > \end{align*}
> >
> > If $a>2$, then using exact MPC gives a solution of $u_t = -ax_t/2$. If $a > 2$, the resulting closed-loop system is unstable.
> >
> > Also, because this is a strongly convex quadratic problem, the Newton step gives the exact optimal solution.
> >
> > Based on the setup of the paper, we will always have $u_{t,t}^0 = 0$ and $u_{t,t}^1 = -\alpha_t a \bar x_t / 2$ . So, if $\alpha_t \le  1$, the system will become unstable.

---

> > > ### Author Response · Authors · 2021-08-26
> > > **We appreciate the reviewer for re-evaluating the paper and having positive opinion.**
> > >
> > > We appreciate that the reviewer could re-evaluate the paper and have positive opinion this time. Indeed, in this case both exact MPC and RTI have unstable closed-loop systems.
> > >
> > >
> > > However, for RTI, we are more interested in the question that whether RTI iterates could stably track the exact MPC iterates; because RTI is purely based on exact MPC and saves most of computations of exact MPC. In other words, we mostly compare with exact MPC, and the stability of RTI (sometimes) refers to the stability of the tracking error to exact MPC. See [16, Theorem 3.5], [53, Theorem 4.2] for example. In this sense, the existing literature ([10, 35, 50] etc.) has shown that RTI is stable locally. This is also evidently true in your example (since $\alpha_t=1$ locally). Our paper does not address the stability in any sense, which is more related to local analysis. Instead, our paper complements the series of works of RTI by performing global analysis, which is always missed before our paper.
> > >
> > > **In summary**, if exact MPC is not stable, expecting RTI to be stable is too strong to hold. However, the stability of RTI towards exact MPC is something known, and our paper contributes to this research direction.

---

### Official Review · Reviewer_ZaaT · 2021-07-16

**Rating:** 7
**Confidence:** 4

**Summary:**

This
 paper proposes an adaptive extension of a real-time iteration (RTI)
scheme used to solve online non linear optimal control problems, with
time-varying dynamics.

It introduces a parametric merit function, named exact augmented
Lagrangian, which is used to adapt the step-size of each unique Newton
steps involved to solve the succession of MPC sub-problems. Such
step-size follows an Armijo-type condition. Before each back-tracking
line search, the penalty parameters of this merit function are adjusted
according to a well-defined scheme.

Under generic assumptions, authors prove the global convergence of their
 approach. That is, the KKT residual of t-th sub-problem converges to
zero as t, the time horizon, increases. Furthermore, for large t,
penalty parameters of the merit function used are no longer changed.

Finally, the paper proposes two types of numerical experiments. First,
it compares the performances (in terms of speed and precision) of their
approach with the one using a $l_1$ merit function to perform a
backtracking scheme. Secondly, it analyses the robustness of their
assumptions with respect to a quadratic penalty parameter $\mu$, which
is tuned manually in their approach.

Numerical results show indeed that this adaptive RTI has a greater
performance, especially for time-varying problems, and is robust to
$\mu$ parameter choice.

**Limitations And Societal Impact:**

Authors
 provides limitations and extensions to their work, including more
precise constants to be found in theorem 4.4, and several remarks
provided in section 6.

Yet, as developed in the previous section, the scope of assumptions made
 on parameter $c$ must be more investigated numerically. If it appears
that for some problems, a too large $\mu$ compromises global
convergence, then such limitation must be developed consequently.




**Main Review:**

The
 paper is original by the subject it tackles, which is to
propose an algorithm ensured to globally converge when used with an
online non-linear optimal control problem with time-varying dynamics.
This task is indeed hard.

Its theoretical contribution is valuable and of some
significance for the community as it provides a first global
convergence analysis of RTI schemes in this setting.

The paper is well written and organized. Explanations are clear. The
following remarks could enhance the quality and clarity of the
contribution :
  - There should be a reference, or a proof, to the assertion of
equivalence between being a strict local minimizer of the exact
augmented Lagrangian and problem 2 (cf. lines  135 to 137),
  - A reference to the mean value theory and back-tracking line
search would be appreciable (at lines 142-143),
 - The argument for tuning $\mu$ manually (lines 150-151) is not really convincing as other schemes could be used (not involving solving several times the KKT systems). This part should be more developed.
  - It is not clear whether or not for large $t$ the augmented
Lagrangian is ensured to be exact with final $\eta_1$ and $\eta_2$ found
 (cf. theorem 4.4). Is it the case ? If yes, it should be mentioned somewhere, otherwise, does it mean the property is useless for this global convergence analysis ? It would be worth precising it.
   - In line 235, I guess authors want to write "one Newton step"
and not "one step Newton".
   - Simulation setups should be enhanced. 5 independent runs are too little to show results of any statistical significance. The mean
value of at least 1000 trajectories (for each different initial point)
with standard deviations should be provided. The same applies finally to the robustness analysis of the $\mu$ parameter. 5 different parameters with a very little scale difference (from 1 to 5) are not significant to prove your claim. The analysis should be done for larger $\mu$ choices
(at least $\mu = 1000$). Indeed, assumption 4.3 involves a parameter
$c$, which authors "hope" being "small" (line 207). Or, the global
convergence theorem 4.7 assumes $c\leq 1/\kappa^3$, with $\kappa = \mu /
 \gamma_{RH}$. It is hence of some importance to analyze numerically the scope of assumptions imposed for $c$.



**Time Spent Reviewing:**

8

---

> ### Author Response · Authors · 2021-08-05
> **We appreciate the reviewer for positive opinion. All your comments are valuable for further improvement and we will address them properly**
>
> ***1***, Indeed. We will provide a reference. See a proof in [2, Proposition 4.15].
>
> ***2***, Indeed. We will make it clearer. See the explanation of backtracking line search in Section 9.2 (Page 465) in ``Convex Optimization" by Boyd et al. We have also rigorously shown the existence of the stepsize in Theorem 4.5.
>
> ***3***, Indeed. We should say the adaptivity on $\mu$ for our procedure is not very promising, but adaptivity may be desirable for other approaches. We will improve this statement.
>
> ***4***, It is a very nice point. In general, there is no guarantee that stabilized parameters always make augmented Lagrangian exact, since stabilized parameters rely on specific iteration sequence. However, stabilized parameters indeed ensure that we converge to a stationary point of the original problem, not just the stationary point of the augmented Lagrangian. As comparison, the parameters thresholds for exact property ensure that ***all*** stationary points of augmented Lagrangian are stationary points of the original problem, which is a stronger guarantee.
>
> However, it does not mean that exactness is not important for analysis. It is important. We implicitly rely on exactness in While loop (Line 4 of Alg. 1), where we can upper bound KKT residual of the original problem by the inner product between augmented Lagrangian gradient and Newton direction. Intuitively, this means that we can upper bound KKT residual by augmented Lagrangian gradient, which is precisely what exact property tries to say. For inexact penalty functions, e.g. vanilla augmented Lagrangian ($\eta_2=0$), our condition in While loop cannot be imposed, and the convergence to KKT point is not guaranteed.
>
> ***5***, Indeed. We will correct this typo.
>
> ***6***, Indeed. We will enhance simulation by trying more initializations, larger $\mu$, and investigate the sharpness of $c$. In the paper, we deliberately use large variance in random initialization to convince readers that the procedure converges for all five initializations. We neglect statistical significance. Thanks for the advice.

---

### Official Review · Reviewer_3owW · 2021-07-20

**Rating:** 6
**Confidence:** 3

**Summary:**

This paper studies a real-time iteration scheme for solving nonlinear MPC. The paper provides an adaptive way of Newton stepsize selection and establishes global convergence. The paper also provides abundant numerical results.

**Limitations And Societal Impact:**

Yes

**Main Review:**

The paper is well-written and the results are important. One major challenge in nonlinear control is the computation burden and one major strength is that the proposed algorithm can ensure global convergence for time-varying cost functions and dynamical systems only by one iteration of Newton update per time step, which greatly reduces the computation time per stage while still ensuring certain performance guarantee. Some comments are provided below.

Firstly,  It would be better if the authors can comment more on the quality of the stationary points where the RTI converges. For example, how does the converged stationary point compare with solving MPC completely? Is there any way to improve the performance of the stationary points? How do the initial values affect the quality of the stationary points?

Secondly, can the authors provide more discussions on the technical novelty? Right now, the results seem to be a straightforward application of line search plus online Newton updates.

Further, the stability of NMPC by implementing the RTI scheme proposed in this paper should be discussed.

Lastly, can the authors comment on how to extend this to the model-free case?

**Time Spent Reviewing:**

2

---

> ### Author Response · Authors · 2021-08-05
> **We thank the reviewer for positive opinion. All your comments are great points and we will incorporate them in the revision**
>
> ***1***, The quality of stationary point is a great point.
>
> **(1)** Indeed, we are not selective to stationary points; it is possible that we converge to saddle point instead of local minimum. However, this is confined to our global assumption. In nonlinear optimization literature, the global result only ensures the convergence to a stationary point (see e.g. [37, Theorem 3.8]). This is still an important first-step result, because stationarity is the first-order necessary condition. Further, under extra local assumptions, we are then able to converge to a local minimum and establish the local convergence rate. This paper focuses on the first part only; the second part has been investigated for RTI.
>
> In particular, our A 4.2 only assumes the boundedness on the modified Hessian $B^t$, and $B^t = \mu I$ is enough for our analysis. If we further assume the modification vanishes $\|\|B^t - H^\star\|\| = o(1)$ (where $H^\star$ is the Hessian at the stationary point, and such condition is commonly assumed in local analysis [37, Theorem 18.7]), then we can show the stationary point that we converge to is a strict local minimum.
>
> **(2)** We believe exact NMPC is a different style and not better than RTI. The exact NMPC requires global convergence for each subproblem. Without further assumptions, exact NMPC solves nonlinear subproblems by, still, finding a stationary point in general due to nonlinearity. If we require NMPC to find local minimum for each subproblem, this may overdo the MPC task because subproblems vary with time stages frequently, and local minimum of the current subproblem does not provide much information to local minimum of the next subproblem. For some special cases (e.g. convex problems or LQR), there is one stationary point for each subproblem, which is local minimum. Then both exact MPC and our RTI will converge to that point, while we has significant less computations.
>
> **(3)** Theoretically, we suspect the vanilla Newton step can be selective to stationary points. However, considering a stochastic algorithm by introducing a random perturbation to iterates may be a way to get rid of saddle points. This is outside the scope the paper though, and existing local analysis has to be adjusted as well.
>
> **(4)** We think that there is no good principle on selecting stationary points based on initial values, especially under generic nonlinear setup. A practical procedure may choose several random initializations, compute Hessian at stationary points, and select them by checking second-order conditions first, and then comparing their objective values.
>
> ***2***, Indeed. We will improve the novelty explanation in the paper. The novelty has several aspects.
>
> **(1)** The problem is open: The known result is that, if initial value is close to the subproblem solution, then RTI stably tracks the subproblem solution. We complement such a local result by showing RTI converges globally, if we incorporate line search for each Newton step.
>
> **(2)** The line search we use for global convergence is indeed standard. However, using differential exact augmented Lagrangian in line search is novel in RTI, and achieve better performance especially for time-varying systems.
>
> **(3)** Our analysis is challenging, and beyond "line search plus online Newton updates". The main difficulty is that problem varies with iteration. For online Newton/gradient updates, the problem is fixed and a new Newton/gradient direction targeting the same problem is provided. However, in time-varying regime, the problem varies with time. Thus, the technical novelty is to investigate how two successive subproblems are related so that the decrease of each subproblem is accumulative. This key step also relies on the used augmented Lagrangian, which may not hold for other merit functions. We also allow to adaptively select penalty parameters for practical purpose.
>
> ***3***, We will improve the discussion on this point, though the paper mainly focus on global behavior. Local stability of RTI has been initially studied in [10] and, recently in [26, 30, 35]. Our procedure reduces to standard RTI scheme when a unit stepsize is accepted locally; so those local results still stand for our procedure (though under additional local assumptions).
>
> ***4***, This is an important future research. The extension to model-based case is relatively easy, though a rigorous analysis including statistical error for model estimation is required. For model-free case, the situation is complex and we are not sure if there is a doable extension. The primary component of model-free case is (nonparametric) learning, while our procedure relies on a parametrization of the model. In principle, we can still ask the question: whether the learned behavior based on the past experience is asymptotically optimal if the environment varies in some way with the learning procedure. This abstract question has the same flavor as our paper, but technical tools would be very different.

---

### Decision · Program_Chairs · 2021-09-27

**Decision:**

Accept (Poster)

**Comment:**

Dear authors,

The authors reached a consensus and positively evaluated the paper. Hence I recommend acceptation to the SAC.
I encourage you to pay special care to clarifying the points raised during the discussion regarding stability, MPC and RTI when preparing the final version of the paper.